# IN VIVO CELL-TYPE AND BRAIN REGION CLASSIFICATION VIA MULTIMODAL CONTRASTIVE LEARNING

**Han Yu[1], Hanrui Lyu[2], Ethan Yixun Xu[1], Charlie Windolf[1], Eric Kenji Lee[3], Fan Yang[4], Andrew M. Shelton[5], Olivier Winter[6], International Brain Laboratory, Eva L. Dyer[7], Chandramouli Chandrasekaran[3], Nicholas A. Steinmetz[8], Liam Paninski[1], Cole Hurwitz[1]**

[1]Columbia University, New York, NY, USA
[2]Northwestern University, Evanston, IL, USA
[3]Boston University, Boston, MA, USA
[4]University College London, London, UK
[5]Allen Institute, Seattle, WA, USA
[6]Champalimaud Foundation, Lisbon, Portugal
[7]Georgia Institute of Technology, Atlanta, GA, USA
[8]University of Washington, Seattle, WA, USA
`{hy2562,yx2740,ciw2107,lmp2107,ch3676}@columbia.edu`
`hanruilyu2029@u.northwestern.edu`
`{kenjilee,cchandr1}@bu.edu`
`{yfan7809,andrew.shelton1312, nick.steinmetz}@gmail.com`
`olivier.winter@internationalbrainlab.org`
`evadyer@gatech.edu`

## ABSTRACT

Current electrophysiological approaches can track the activity of many neurons, yet it is usually unknown which cell-types or brain areas are being recorded without further molecular or histological analysis. Developing accurate and scalable algorithms for identifying the cell-type and brain region of recorded neurons is thus crucial for improving our understanding of neural computation. In this work, we develop a multimodal contrastive learning approach for neural data that can be fine-tuned for different downstream tasks, including inference of cell-type and brain location. We utilize this approach to jointly embed the activity autocorrelations and extracellular waveforms of individual neurons. We demonstrate that our embedding approach, **N**euronal **E**mbeddings via **M**ultim**O**dal contrastive learning (NEMO), paired with supervised fine-tuning, achieves state-of-the-art cell-type classification for two opto-tagged datasets and brain region classification for the public International Brain Laboratory Brain-wide Map dataset. Our method represents a promising step towards accurate cell-type and brain region classification from electrophysiological recordings. The project page and code are available at `https://ibl-nemo.github.io/`.

## 1 INTRODUCTION

High-density electrode arrays now allow for simultaneous extracellular recording from hundreds to thousands of neurons across interconnected brain regions (Jun et al., 2017; Steinmetz et al., 2021; Ye et al., 2023b; Trautmann et al., 2023). While significant progress has been made in developing algorithms for tracking neural activity (Hennig et al., 2019; Buccino et al., 2020; Magland et al., 2020; Boussard et al., 2023; Pachitariu et al., 2024), identifying cell types and brain regions solely from electrophysiological features remains an open problem.

Traditional approaches for electrophysiological cell-type classification utilize simple features of the extracellular action potential (EAP) such as its width or peak-to-trough amplitude (Mountcastle et al., 1969; Matthews & Lee, 1991; Nowak et al., 2003; Barthó et al., 2004; Vigneswaran et al., 2011) or features of neural activity, such as the inter-spike interval distribution (Latuske et al., 2015; Jouty et al., 2018). These simple features are interpretable and easy to visualize but lack discriminative power and robustness across different datasets (Weir et al., 2015; Gouwens et al., 2019). Current

automated featurization methods for EAPs (Lee et al., 2021; Vishnubhotla et al., 2024) and neural activity (Schneider et al., 2023a) improve upon manual features but are limited to a single modality.

There has been a recent push to develop multimodal methods that can integrate information from both recorded EAPs and spiking activity. PhysMAP (Lee et al., 2024) is a UMAP-based (McInnes et al., 2018a) approach that can predict cell-types using multiple physiological modalities through a weighted nearest neighbor graph. Another recently introduced method utilizes variational autoencoders (VAEs) to embed each physiological modality separately and then combines these embeddings before classification (Beau et al., 2025). Although both methods show promising results, PhysMAP is hard to fine-tune for downstream tasks as it is nondifferentiable, and the VAE-based method captures features that are important for reconstruction, not discrimination, impairing downstream performance (Guo et al., 2017). Neither approach has been used to classify brain regions.

In this work, we introduce a multimodal contrastive learning method for neurophysiological data, Neuronal Embeddings via MultimOdal Contrastive Learning (NEMO), which utilizes large amounts of unlabeled paired data for pre-training and can be fine-tuned for different downstream tasks including cell-type and brain region classification. We utilize a recently developed contrastive learning framework (Radford et al., 2021) to jointly embed individual neurons' activity autocorrelations and average extracellular waveforms. The key assumption of our method is that jointly embedding different modalities into a shared latent space will capture shared information while discarding modality-specific noise (Huang et al., 2024). We evaluate NEMO on cell-type classification using optotagged Neuropixels Ultra (NP Ultra) data from the mouse visual cortex (Ye et al., 2023b) and optotagged Neuropixels 1 data from the mouse cerebellum (Beau et al., 2025). We evaluate NEMO on brain region classification using the International Brain Laboratory (IBL) Brain-wide Map dataset (IBL et al., 2023). Across all datasets and tasks, NEMO outperforms current unsupervised (PhysMAP and VAEs) and supervised methods, with particularly strong performance in label-limited regimes. These results demonstrate that NEMO is a significant advance towards accurate cell-type and brain region classification from electrophysiological recordings.

## 2 RELATED WORK

### 2.1 CONTRASTIVE LEARNING FOR NEURONAL DATASETS

The goal of contrastive learning is to find an embedding space where similar examples are close together while dissimilar ones are well-separated (Le-Khac et al., 2020). Contrastive learning has found success across a number of domains including language (Reimers & Gurevych, 2019), vision (Chen et al., 2020), audio (Saeed et al., 2021), and multimodal learning (Radford et al., 2021; Tian et al., 2020). Contrastive learning has also been applied to neuronal morphological data (Chen et al., 2022), connectomics data (Dorkenwald et al., 2023) and preprocessed spiking activity (Azabou et al., 2021; Urzay et al., 2023; Schneider et al., 2023b; Antoniades et al., 2023). In each of these applications, associated downstream tasks such as 3D neuron reconstruction, cellular sub-compartment classification, or behavior prediction have shown improvement using this contrastive paradigm. One contrastive method, CEED, has been applied to raw extracellular recordings to perform spike sorting and cell-type classification. In contrast to NEMO, CEED is unimodal (it ignores neural activity) and has never been applied to optotagged data or brain region classification (Vishnubhotla et al., 2024).

### 2.2 CELL-TYPE CLASSIFICATION

The goal of cell-type classification is to assign neurons to distinct classes based on their morphology, function, electrophysiological properties, and molecular markers (Masland, 2004). Current transcriptomic (Tasic et al., 2018; Gala et al., 2019; Yao et al., 2021; 2023) and optical methods (Cardin et al., 2010; Kravitz et al., 2013; Lee et al., 2020) are effective but require extensive sample preparation or specialized equipment, limiting their scalability and applicability for in-vivo studies (Lee et al., 2024). Recently, calcium imaging has been utilized in conjunction with molecular approaches to identify cell-types (Bugeon et al., 2022; Mi et al., 2023). This approach has low temporal resolution and requires substantial post hoc effort to align molecular imaging with calcium data, making it unsuitable for closed-loop in vivo experiments. A promising alternative is to use the electrophysiological properties of recorded neurons as they capture some of the variability of the transcriptomic profile (Bomkamp et al., 2019). Simple electrophysiological features from a neuron's EAP and spiking activity are commonly used to identify putative cell types, such as excitatory and inhibitory cells (Frank et al., 2001; Gouwens et al., 2019). Automated methods including

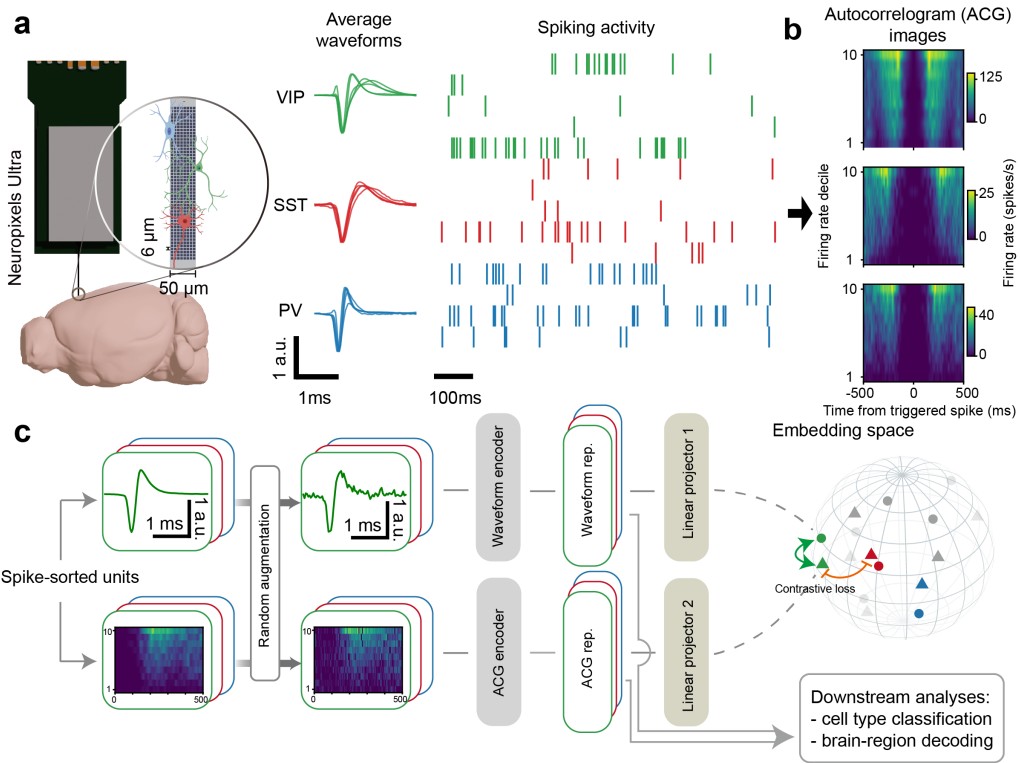

Figure 1: **Schematic illustration of NEMO.** (a) Neuropixels Ultra (Ye et al., 2023b) recordings capture activity from many different cell-types which have distinct extracellular action potentials (EAPs) and spiking activity. We present waveform and spiking activity snippets from five example neurons for each cell-type. (b) We transform the spiking activity of each neuron into a compact autocorrelogram (ACG) image from (Beau et al., 2025) that accounts for variations in the firing rate (see Section 4.1) (c) NEMO utilizes a CLIP-based objective where an EAP encoder and an ACG image encoder are trained to embed randomly augmented EAPs and ACG image from the same neuron close together while keeping different neurons separate. The learned representations can be utilized for downstream tasks such as cell-type and brain-region classification.

EAP-specific methods (Lee et al., 2021) and activity-based methods (Schneider et al., 2023a) are an improvement in comparison to manual features. Most recently, multi-modal cell-type classification methods including PhysMAP (Lee et al., 2024) and a VAE-based algorithm (Beau et al., 2025; Herzfeld et al., 2025) have been introduced which make use of multiple physiological modalities such as the EAP, activity, or peri-stimulus time histogram (PSTH).

### 2.3 BRAIN REGION CLASSIFICATION

Brain region classification consists of predicting the location of a neuron or electrode based on the recorded physiological features (Steinmetz et al., 2018; Davis et al., 2023). Rather than predicting a 3D location, the task is to classify the brain region a neuron or electrode occupies, which can be estimated using post-insertion localization via histology (Sunkin et al., 2012). Brain region classification is an important task for understanding fundamental differences in physiology between brain areas and for targeting regions that are hard to hit via insertion. Most importantly, brain region classification can provide a real-time estimate of the probe's location in the brain during experiments. Additionally, insertions in primates and human subjects often lack histological information, instead relying on the experimental heuristics that lack standardization between laboratories. As this task is relatively new, only simple features of the EAPs have been utilized for classification (Jia et al., 2019; Tolossa et al., 2024).

### 3 DATASETS

For cell-type classification, we use two mouse datasets: an opto-tagged dataset from the visual cortex recorded with Neuropixels Ultra (NP Ultra; Ye et al., 2023b) and a dataset from the cerebellar cortex

recorded with Neuropixels 1 (Beau et al., 2025). For brain region classification, we utilize the IBL Brain-wide Map of neural activity from mice performing a decision-making task (IBL et al., 2023).

**NP Ultra opto-tagged mouse data.** This dataset consists of NP Ultra recordings of spontaneous and opto-stimulated activity from the visual cortex of mice. We included spontaneous periods only and excluded units that have less than 100 spikes after removal of stimulation periods (see Supplement C). We obtained 462 ground-truth neurons with three distinct cell-types. The ground-truth neurons are composed of 237 parvalbumin (PV), 107 somatostatin (SST), and 118 vasoactive intestinal peptide cells (VIP). There are also 8491 unlabelled neurons that we can utilize for pre-training.

**C4 cerebellum dataset.** This dataset consists of Neuropixels recordings from the cerebellar cortex of mice. Opto-tagging is utilized to label 202 ground-truth neurons with five distinct cell-types. The ground-truth neurons are composed of 27 molecular layer interneurons (MLI), 18 Golgi cells (GoC), 30 mossy fibers (MF), 69 Purkinje cell simple spikes (PCss), and 58 Purkinje cell complex spikes (PCcs). There are 3,090 unlabelled neurons for pretraining.

**IBL Brain-wide Map.** This dataset consists of Neuropixels recordings from animals performing a decision-making task. Each neuron is annotated with the brain region where it is located. We utilize 675 insertions from over 100 animals, yielding 37017 'good' quality neurons after spike sorting and quality control (Banga et al., 2022; IBL et al., 2022). Each brain is parcellated with 10 brain atlas annotations divided into 10 broad areas: isocortex, olfactory areas (OLF), cortical subplate (CTXsp), cerebral nuclei (CNU), thalamus (TH), hypothalamus (HY), midbrain (MB), hindbrain (HB), cerebellum (CB) and hippocampal formation (HPF). We divide this dataset into a training, validation, and testing set by insertion such that we can evaluate each model on heldout insertions.

## 4 NEMO

We introduce **N**euronal **E**mbeddings via **M**ultim**O**dal contrastive learning (NEMO), which learns a multimodal embedding of neurophysiological data. To extract representations from multiple modalities, we utilize Contrastive Language-Image Pretraining (CLIP; Radford et al., 2021). CLIP uses a contrastive objective to learn a joint representation of images and captions. For NEMO, we utilize the same objective but with modality-specific data augmentations and encoders (see Figure 1c).

### 4.1 PREPROCESSING

We construct a paired dataset of spiking activity and EAPs for all recorded neurons. Using the open-source Python package NeuroPyxels (Beau et al., 2021), we computed an autocorrelogram (ACG) image for each neuron by smoothing the spiking activity with a 250-ms width boxcar filter, dividing the firing rate distribution into 10 deciles, and then building ACGs for each decile (see Figure 1b). This ACG image is a useful representation because the activity autocorrelations of a neuron can change as a function of its firing rate. By computing ACGs for each firing rate decile, the ACG image will account for firing rate dependent variations in the autocorrelations, allowing for comparisons between different areas of the brain, behavioral contexts, and animals (Beau et al., 2025). For the EAPs, we construct a 'template' waveform which is the mean of ∼500 waveforms for that neuron. For NP Ultra, we utilize multi-channel templates which take advantage of the detailed spatial structure enabled by the small channel spacing; we use nine channels with the highest peak-to-peak (ptp) amplitude, re-ordered from highest to lowest amplitude. For the C4 and IBL dataset, all main text results utilize templates consisting of one channel with maximal amplitude.

### 4.2 DATA AUGMENTATIONS

Previous work on contrastive learning for spiking activity utilizes data augmentations including sparse multiplicative noise (pepper noise), Gaussian noise, and temporal jitter (Azabou et al., 2021). As it is computationally expensive to construct ACG images for each batch during training, we instead design augmentations directly for the ACG images rather than the original spiking data. Our augmentations include temporal Gaussian smoothing, temporal jitter, amplitude scaling, additive Gaussian noise, and multiplicative pepper noise (see Supplemental B for more details and Supplementary Figure 14 for an ablation). For single channel templates, we use additive Gaussian noise as our only augmentation. For multi-channel templates, we also include electrode dropout and amplitude jitter as described in Supplementary Table 1.

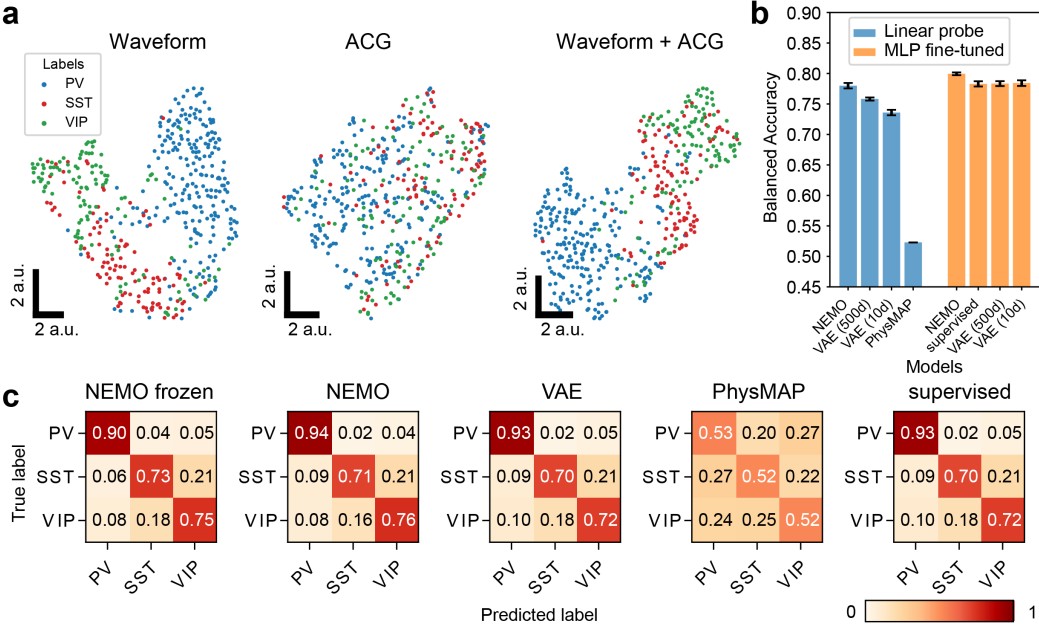

Figure 2: **Comparing NEMO to baseline models on the NP Ultra opto-tagged dataset.** (a) UMAP visualization of the pretrained NEMO representations of unseen opto-tagged visual cortex units, colored by different cell-types. Neurons of the same class form clusters, particularly when combined modalities are used. (b) Balanced accuracy and (c) Confusion matrices for the NP Ultra classification results, normalized by ground truth label and averaged across 5 random seeds. NEMO outperforms the other embedding methods by a significant margin across all cell-types and evaluation methods.

## 4.3 ENCODERS

We employ separate encoders for each electrophysiological modality. For the ACG image encoder, we use a version of the convolutional architecture introduced in (Beau et al., 2025) with 2 layers and Gaussian Error Linear Units (GeLU) (Hendrycks & Gimpel, 2016). For the waveform encoder, we use a 2 layer multilayer perceptron (MLP) with GeLU units. The representation sizes are 200 dimensional and 300 dimensional for the ACG image encoder and the waveform encoder, respectively. We set the projection size to be 512. For details about hyperparameters, see Supplement B.

## 4.4 CONTRASTIVE OBJECTIVE

We utilize the contrastive objective defined in CLIP. Let $z_{\text{acg}}$ and $z_{\text{wf}}$ be the L2 normalized projections of each modality. For a batch $B$, the objective is as follows,

$$\mathcal{L} = -\frac{1}{2|B|} \sum_{i=1}^{|B|} \left[ \log \frac{\exp(z_{\text{acg}_i} \cdot z_{\text{wf}_i}/\tau)}{\sum_{j=1}^{|B|} \exp(z_{\text{acg}_i} \cdot z_{\text{wf}_j}/\tau)} + \log \frac{\exp(z_{\text{acg}_i} \cdot z_{\text{wf}_i}/\tau)}{\sum_{j=1}^{|B|} \exp(z_{\text{acg}_j} \cdot z_{\text{wf}_i}/\tau)} \right] \quad (1)$$

where $\tau$ is a temperature parameter which we fix during training. The objective function encourages the model to correctly match $z_{\text{acg}_i}$ with its corresponding $z_{\text{wf}_i}$, and vice versa, over all other possible pairs in the batch. This loss can easily be extended to more than two modalities including PSTHs.

## 4.5 SINGLE-NEURON AND MULTI-NEURON BRAIN REGION CLASSIFICATION

Brain region classification using electrophysiological datasets is a new problem that requires novel classification schemes. We develop two classification schemes for our evaluation: a single-neuron and multi-neuron classifier. For our single-neuron classifier, we predict the brain area for each neuron independently using its embedding. For our multi-neuron classifier, we predict the brain region for each 20 micron bin along the depth of the probe by ensembling the predictions of nearby neurons within a 60-micron radius (i.e., averaging the logits of the single-neuron model) as shown in Figure 3a (ii). When more than five neurons fall within this range, only the nearest five are selected.

Table 1: **Cell-type classification for the NP Ultra dataset.** 5-Fold accuracy and F1-scores are reported for three conditions: (i) a linear layer and (ii) MLP on top of the frozen pretrained representations, and (iii) after MLP finetuning. Chance level is 0.33 for this dataset.

| Model | Linear | | MLP | | MLP fine-tuned | |
|---|---|---|---|---|---|---|
| | Acc | F1 | Acc | F1 | Acc | F1 |
| Supervised | N/A | N/A | N/A | N/A | $0.79 \pm 0.00$ | $0.78 \pm 0.00$ |
| PhysMAP (Lee et al. (2024)) | $0.52^1$ | $0.52^1$ | N/A | N/A | N/A | N/A |
| VAE (10D latent; Beau et al. (2025)) | $0.74 \pm 0.00$ | $0.73 \pm 0.00$ | $0.74 \pm 0.00$ | $0.73 \pm 0.00$ | $0.78 \pm 0.00$ | $0.79 \pm 0.00$ |
| VAE (500D rep) | $0.76 \pm 0.00$ | $0.75 \pm 0.00$ | $0.77 \pm 0.00$ | $0.77 \pm 0.00$ | $0.78 \pm 0.00$ | $0.79 \pm 0.00$ |
| NEMO (500D rep) | $\mathbf{0.78 \pm 0.01}$ | $\mathbf{0.78 \pm 0.00}$ | $\mathbf{0.80 \pm 0.00}$ | $\mathbf{0.79 \pm 0.00}$ | $\mathbf{0.80 \pm 0.00}$ | $\mathbf{0.80 \pm 0.00}$ |

Table 2: **Cell-type classification for the C4 cerebellum dataset.** 5-Fold accuracy and F1-scores are reported for three conditions: (i) a linear layer and (ii) MLP on top of the frozen pretrained representations, and (iii) after MLP finetuning. Chance level is 0.20 for this dataset.

| Model | Linear | | MLP (5-fold) | | Finetuned MLP (5-fold) | |
|---|---|---|---|---|---|---|
| | Acc | F1 | Acc | F1 | Acc | F1 |
| Supervised | N/A | N/A | N/A | N/A | $0.82 \pm 0.00$ | $0.82 \pm 0.00$ |
| PhysMAP | $0.51^1$ | $0.49^1$ | N/A | N/A | N/A | N/A |
| VAE (10D latent; Beau et al. (2025)) | $0.79 \pm 0.00$ | $0.78 \pm 0.01$ | $0.74 \pm 0.01$ | $0.73 \pm 0.02$ | $0.82 \pm 0.00$ | $0.81 \pm 0.00$ |
| VAE (500D rep) | $0.82 \pm 0.00$ | $0.81 \pm 0.00$ | $0.82 \pm 0.00$ | $0.82 \pm 0.00$ | $0.83 \pm 0.00$ | $0.83 \pm 0.00$ |
| NEMO (500D rep) | $\mathbf{0.85 \pm 0.01}$ | $\mathbf{0.85 \pm 0.01}$ | $\mathbf{0.85 \pm 0.00}$ | $\mathbf{0.85 \pm 0.00}$ | $\mathbf{0.86 \pm 0.01}$ | $\mathbf{0.86 \pm 0.01}$ |

## 5 EXPERIMENTAL SETUP

### 5.1 BASELINES

For our baselines, we compare to PhysMAP (Lee et al., 2024), a VAE-based method (Beau et al., 2025)), and a fully supervised MLP. For fair comparison, we utilize the same encoder architectures for NEMO and the VAE-based method. We include two versions of the VAE baseline: (1) the latent space (10D) is used to predict cell-type or brain region (from Beau et al. (2025)), or (2) the output of the layer before the latent space (500D) is used to predict cell-type or brain region. Although this approach was not proposed in Beau et al. (2025), we find that utilizing the 500D representations *before* the latent space performed better than using the 10D latent space and also outperformed a VAE trained with a 500D latent space. For the VAEs, we use default hyperparameters from Beau et al. (2025) (see Supplement N for a hyperparameter sensitivity analysis). For the supervised baseline, we again use the same encoder architectures as NEMO. For training NEMO, we use an early stopping strategy which utilizes validation data. For the VAE-based method, we use the training scheme introduced in Beau et al. (2025). We fix the hyperparameters for all methods across all datasets. For more details about baselines, training, and hyperparameters, see Supplements B and D.

### 5.2 EVALUATION

For both NEMO and the VAE-based method, the representations from the ACG image and EAPs are concatenated together before classification or fine-tuning. We apply three classification schemes for evaluation including (1) freezing the model and training a linear classifier on the final-layer outputs, (2) freezing the model and training a MLP-based classifier on the final-layer outputs, (3) fine-tuning both the original model and a MLP-based classifier on the final layer. To ensure balanced training data, we implement dataset resampling prior to fitting the linear classifier. For PhysMAP comparisons, we utilize the weighted graph alignment approach provided in Lee et al. (2024) for all comparisons. For our classification metrics, we utilize the macro-averaged F1 score, calculated as the unweighted mean of F1 scores for each class, and balanced accuracy, which measures average accuracy per class. For additional details about baseline hyperparameters, see Supplement B.

### 5.3 EXPERIMENTS

**NP Ultra opto-tagged dataset.** For the NP Ultra dataset, we pretrain NEMO and the VAE-based method on 8491 unlabelled neurons. This pretraining strategy is important for reducing overfitting to the small quantity of labeled cell-types. To evaluate each model after pretraining, we perform the three evaluation schemes introduced in Section 5.2: freezing + linear classifier, freezing + MLP

---

[1] We utilize PhysMAP's anchor alignment technique (which is deterministic) to evaluate its performance.

Table 3: **Single-neuron brain region classification for the IBL dataset.** The accuracy and F1-scores are reported for three conditions: (i) a linear layer and (ii) MLP on top of the frozen pretrained representations, and (iii) after MLP finetuning. Chance level is 0.10 for this dataset.

| Model | Linear | | MLP | | MLP fine-tuned | |
|---|---|---|---|---|---|---|
| | Acc | F1 | Acc | F1 | Acc | F1 |
| Supervised | N/A | N/A | N/A | N/A | $0.45 \pm 0.01$ | $0.42 \pm 0.01$ |
| PhysMAP | $0.31^1$ | $0.27^1$ | N/A | N/A | N/A | N/A |
| VAE (500D rep) | $0.40 \pm 0.01$ | $0.37 \pm 0.01$ | $0.41 \pm 0.00$ | $0.37 \pm 0.00$ | $0.45 \pm 0.01$ | $0.43 \pm 0.00$ |
| NEMO (500D rep) | $\mathbf{0.42 \pm 0.00}$ | $\mathbf{0.40 \pm 0.00}$ | $\mathbf{0.45 \pm 0.01}$ | $\mathbf{0.42 \pm 0.00}$ | $\mathbf{0.47 \pm 0.01}$ | $\mathbf{0.44 \pm 0.01}$ |

Table 4: **Multi-neuron brain region classification for the IBL dataset.** The accuracy and F1-scores are reported for three conditions: (i) a linear layer and (ii) MLP on top of the frozen pretrained representations, and (iii) after MLP finetuning. Chance level is 0.10 for this dataset.

| Model | Linear | | MLP | | MLP fine-tuned | |
|---|---|---|---|---|---|---|
| | Acc | F1 | Acc | F1 | Acc | F1 |
| Supervised | N/A | N/A | N/A | N/A | $0.50 \pm 0.00$ | $0.48 \pm 0.01$ |
| VAE (500D rep) | $0.45 \pm 0.01$ | $0.42 \pm 0.01$ | $0.46 \pm 0.00$ | $0.43 \pm 0.00$ | $0.51 \pm 0.01$ | $0.49 \pm 0.01$ |
| NEMO (500D rep) | $\mathbf{0.48 \pm 0.00}$ | $\mathbf{0.45 \pm 0.00}$ | $\mathbf{0.50 \pm 0.00}$ | $\mathbf{0.48 \pm 0.00}$ | $\mathbf{0.51 \pm 0.00}$ | $\mathbf{0.50 \pm 0.00}$ |

classifier, and full end-to-end finetuning. For PhysMAP, we utilize the anchor alignment technique introduced by Lee et al. (2024). For all methods and evaluation schemes, we perform a 5-fold cross-validation with 10 repeats to evaluate each model.

**C4 cerebellum dataset.** For the C4 dataset, we pretrain all methods on 3090 unlabelled neurons. To evaluate each model after pretraining, we perform the three evaluation schemes introduced in Section 5.2. For all classifiers, we do not utilize input layer information, nor do we exclude neurons based on a confidence threshold as done in Beau et al. (2025), as we were interested in evaluating the predictiveness of the features directly without additional information. For all methods and evaluation schemes, we perform a 5-fold cross-validation with 10 repeats to evaluate each model.

**IBL Brain-wide Map.** For the IBL dataset, we randomly divide all insertions (i.e., Neuropixels recordings) into a 70-10-20 split to create a training, validation, and test set for each method. We then pretrain NEMO and the VAE-based method on all neurons in the training split. We then perform the three evaluation schemes introduced in Section 5.2. For PhysMAP, we utilize the anchor alignment technique. We train both a single-neuron and multi-neuron classifier using the representations learned by NEMO and the VAE-based method. For PhysMAP, we only evaluate the single-neuron classifier. We compute the average and standard deviation of the metrics using five random seeds.

# 6 RESULTS

## 6.1 CLASSIFICATION

**NP Ultra cell-type classifier.** The results for the NP Ultra opto-tagged dataset are shown in Table 1 and Figure 2. For all three evaluation schemes, NEMO achieves the highest macro-averaged F1 score and balanced accuracy by a significant margin. Notably, its largest improvement is in the linear and frozen MLP evaluations, indicating that NEMO captures cell-type-discriminative features even without fine-tuning. After fine-tuning, all methods are closer in performance, suggesting that there is some saturation for this dataset. These results demonstrate that NEMO is an accurate method for cell-type classification in visual cortical microcircuits even without fine-tuning.

**C4 cerebellum cell-type classifier.** The results for the C4 cerebellum dataset are shown in Table 2 and Supplementary Figure 4. For all evaluation schemes, NEMO outperforms the baseline models, achieving the highest macro-averaged F1 score and balanced accuracy, with the largest gains again in the linear and frozen MLP evaluations. These findings demonstrate that NEMO can accurately classify cell types in the cerebellum without any hyperparameter adjustments.

**IBL single-neuron region classifier.** We then aim to investigate how much relevant information NEMO extracts from each neuron about its anatomical location, i.e., brain region. We investigate this by training classifiers that use single neuron features to identify anatomical regions for the

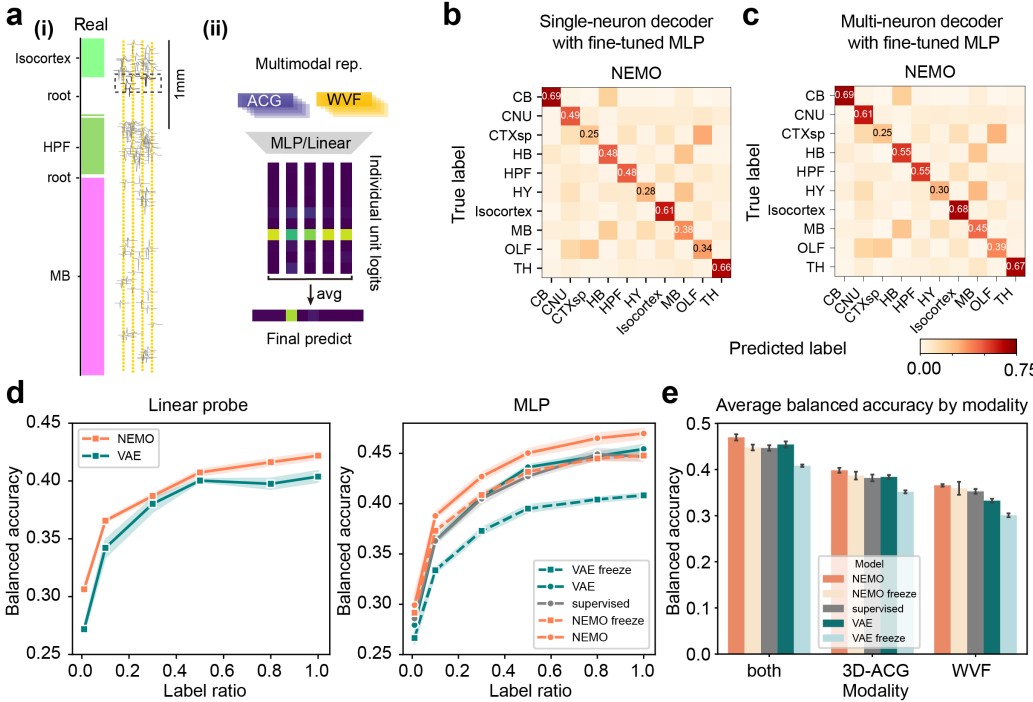

Figure 3: **Results for NEMO on the IBL brain region classification task.** (a) Schematic for multi-neuron classifier. (i) At each depth, the neurons within 60 μm were used to classify the anatomical region. Only the nearest 5 neurons were selected if there were more than 5 neurons within that range. (ii) For logits averaging, single-neuron classifier logits are predicted using a linear model/MLP trained on the representations of our two physiological modalities. The final prediction is based on the average of the individual logits. (b) Confusion matrices for the single-neuron region classifier for fine-tuned NEMO. (c) Confusion matrices for the NEMO multi-neuron region classifier, averaged across 5 runs. (d) Single neuron balanced accuracy with linear classifier and the MLP head for each model trained/fine-tuned with different label ratios. (e) Single-neuron MLP-classification balanced accuracy for each modality separately and for the combined representation.

IBL dataset (see Table 3 and Figure 3 for results). The confusion matrix for the VAE, supervised MLP, and PhysMAP are shown in Supplementary Figure 9. We find that NEMO outperforms all other methods using both the linear and MLP-based classification schemes. Without end-to-end fine-tuning, NEMO with an MLP classification head is already on par with the supervised MLP. NEMO's success with both the linear and MLP classifier with frozen encoder weights indicates that NEMO is able to extract a region-discriminative representation of neurons without additional fine-tuning. This representation can be further improved by fine-tuning NEMO. All methods have closer performance after fine-tuning potentially due to the substantial amount of labeled data.

**IBL multi-neuron region classifier.** We investigate whether combining information from multiple neurons at each location can improve brain region classification. We use the nearest-neurons ensembling approach as described in 4.5 and shown in Figure 3a. Averaging the logits of predictions from single neurons improves classification performance over the single-neuron model. NEMO still has the best region classification performance especially for the linear and frozen MLP evaluations (see Table 4 and Figure 3c for results). Again, all methods have closer performance after fine-tuning.

## 6.2 CLUSTERING

We next examine the clusterability of NEMO representations for the IBL Brain-wide Map. We followed the clustering strategy used in Lee et al. (2021) by running Louvain clustering on a UMAP graph constructed from the representations extracted by NEMO from the IBL training neurons. We adjusted two main settings: the neighborhood size in UMAP and the resolution in Louvain clustering. We selected these parameters by maximizing the modularity index, which had the effect of minimizing the number of resulting clusters (Figure 4c). The clustering results relative to the region labels are presented in Figures 4a and b. The UMAP visualization of the NEMO represen-

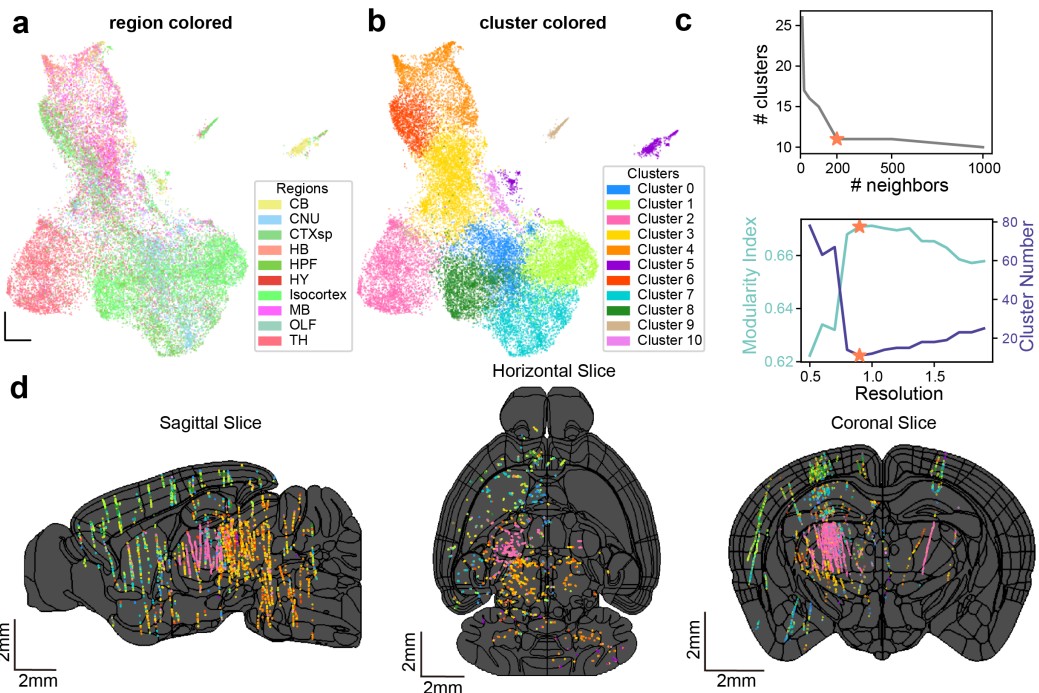

Figure 4: **IBL neuron clustering using NEMO pretraining.** (a) A UMAP visualization of the representations that NEMO extracts from the training data colored by anatomical brain region. (b) The same UMAP as shown in (a) but instead colored by cluster labels using a graph-based approach (Louvain clustering). (c) We tuned the neighborhood size in UMAP and the resolution for the clustering. These parameters were selected by maximizing the modularity index which minimized the number of clusters. (d) 2D brain slices across three brain views with the location of individual neurons colored using the cluster IDs shown in (b). The black lines show the region boundaries of the Allen mouse atlas (Wang et al., 2020). The cluster distribution found using NEMO is closely correlated with the anatomical regions and is consistent across insertions from different labs.

tations, colored by region label, demonstrates that the regions are separable in the representation space. Notably, there is a distinct separation of thalamic neurons from other regions, along with an isolated cluster of cerebellar neurons. Neurons from other regions are also well organized by region labels within the NEMO representation space, allowing for their clustering into several distinct clusters. Additionally, overlaying the neurons colored by their cluster IDs onto their anatomical locations (Figure 4) reveals a cluster distribution closely correlated with anatomical regions which is consistent across insertions from different labs (Supplementary Figure 11). We find that clustering NEMO's representations leads to a more region-selective clustering than when we use the raw features directly (Supplementary Figures 12 and 13). These results demonstrate that NEMO extracts features that capture the electrophysiological diversity across regions even without labels.

## 6.3 ABLATIONS

**Label ratio sweep.** We assess whether NEMO requires less labeled data for fine-tuning by conducting a label ratio sweep with single-neuron region classifiers. We train both linear and MLP classifiers under two conditions: with frozen weights and full end-to-end fine-tuning, using 1%, 10%, 30%, 50%, 80%, and 100% of the labeled data. Accuracy results are shown in Figure 3d (F1 scores in Supplementary Figure 6). The fine-tuned NEMO model outperforms all other methods across all label ratios. Notably, with only 50% of the training labels, both the linear and fine-tuned NEMO models outperform the corresponding VAE models and the supervised MLP, even when the latter models are trained on the full dataset.

**Single modality classifier.** We examine whether combining modalities improves region-relevant information and if NEMO enhances feature extraction by aligning their embeddings. We compared the classification performance of the MLP classifier with encoder weights frozen and end-to-end fine-tuned, across all models using: 1) waveforms only 2) ACGs only 3) waveforms and ACGs.

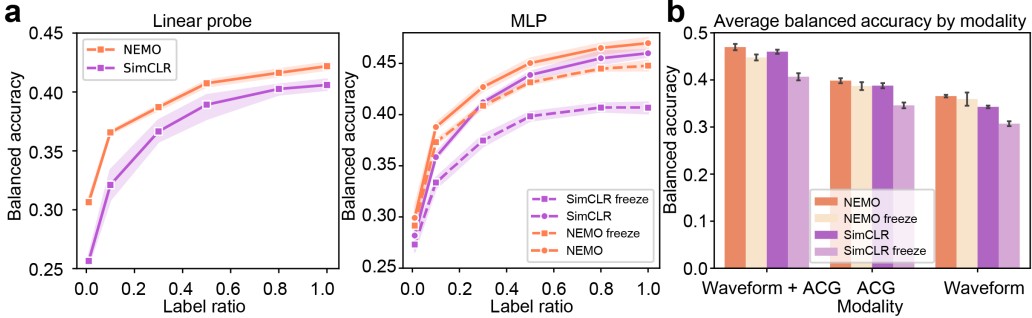

Figure 5: **Ablating joint vs. independent learning for NEMO.** To evaluate the importance of learning a shared representation between modalities, we train a version of NEMO on the IBL brain classification task where each modality is independently embedded using SimCLR. (a) Across all label ratios and classifiers, we find that NEMO trained with CLIP outperforms the SimCLR version. (b) NEMO trained with CLIP also extracts more informative representations for each modality than when training with SimCLR.

Brain region classification balanced accuracies are shown in Figure 3e (for F1, see Supplementary Figure 6). We found that bimodal models generally outperform unimodal models, suggesting that combining both modalities provides extra information on the anatomical location of neurons. Once again, NEMO achieves the best performance, demonstrating its ability to enhance single-modality information extraction by leveraging the other modality. After fine-tuning, NEMO and the VAE have closer performance potentially due to the substantial amount of ground-truth labels.

**Joint vs. independent learning for NEMO.** To ablate the importance of learning a shared representation of each modality, we train a version of NEMO where we independently learn an embedding for each modality using a unimodal contrastive method, SimCLR (Chen et al., 2020). The results for brain region classification are shown in Figure 5a where NEMO trained with CLIP outperforms NEMO trained with SimCLR for all label ratios and classification methods. NEMO trained with CLIP is also able to extract more informative representations from each modality as shown in Figure 5b. These results demonstrate that learning a shared representation of the two modalities is important for the downstream performance of NEMO. Supplementary Table 9 and 10 show that joint training with CLIP also leads to an improvement over SimCLR for cell-type classification.

# 7 DISCUSSION

In this work, we proposed NEMO, a pretraining framework for electrophysiological data that utilizes multi-modal contrastive learning. We demonstrate that NEMO is able to extract informative representations for cell-type and brain region classification with minimal fine-tuning across three different datasets. This is especially valuable in neuroscience, where ground truth data, like opto-tagged cells, are costly, labor-intensive, or even impossible to obtain (e.g., in human datasets).

Our work has several limitations. First, we focus on shared information between two modalities, assuming this is most informative for identifying cell identity or anatomical location. However, modeling both shared and modality-specific information could further improve performance, as each modality may contain unique features relevant to cell identity or anatomical location (Liu et al., 2024; Liang et al., 2024). Additionally, NEMO utilizes the activity of each neuron independently, which ignores neuron correlations that can help distinguish cell-types (Mi et al., 2023). Extending NEMO to encode population-level features is an exciting future direction.

NEMO opens up several promising avenues for future research. Our framework can be adapted for studies of peripheral nervous systems, such as the retina (Wu et al., 2023). NEMO can also be combined with RNA sequencing to find features that are shared between RNA and electrophysiological data (Li et al., 2023). It will also be possible to correlate the cell-types discovered using NEMO with animal behavior to characterize their functional properties. Finally, we imagine that the representations extracted by NEMO can be integrated with current multi-animal pretraining approaches for neural activity to provide additional cell-type information which could improve generalizability to unseen sessions or animals (Azabou et al., 2023; Ye et al., 2023a; Zhang et al., 2025; 2024).

## 8    ACKNOWLEDGMENTS

We thank Jonathan Pillow and Tatiana Engel for providing feedback on this manuscript and Sahar Minavi and Shawn Olsen for their contributions to collecting and sharing the NP Ultra data. We also thank Maxime Beau and the other authors of Beau et al. (2025) for sharing the C4 cerebellum dataset. This project was supported by the Wellcome Trust (PRF 209558, 216324, 201225, and 224688 to MH, SHWF 221674 to LFR, collaborative award 204915 to MC, MH and TDH), National Institutes of Health (1U19NS123716), the Simons Foundation, the DoD OUSD (R&E) under Cooperative Agreement PHY-2229929 (The NSF AI Institute for Artificial and Natural Intelligence), the Kavli Foundation, the Gatsby Charitable Foundation (GAT3708), the NIH BRAIN Initiative (U01NS113252 to NAS, SRO, and TDH), the Pew Biomedical Scholars Program (NAS), the Max Planck Society (GL), the European Research Council under the European Union's Horizon 2020 research and innovation programme (grant agreement No 834446 to GL and AdG 695709 to MH), the Giovanni Armenise Harvard Foundation (CDA to LFR), the Human Technopole (ECF to LFR), the NSF (IOS 211500 to NBS), the Klingenstein-Simons Fellowship in Neuroscience (NAS), the NINDS R01NS122969, the NINDS R21NS135361, the NINDS F31NS131018, the NSF CAREER awards IIS-2146072, as well as generous gifts from the McKnight Foundation, and the CIFAR Azrieli Global Scholars Program. GM is supported by a Boehringer Ingelheim Fonds PhD Fellowship. The primate research procedures were supported by the NIH P51 (OD010425) to the WaNPRC, and animal breeding was supported by NIH U42 (OD011123). Computational modeling work was supported by the European Union Horizon 2020 Research and Innovation Programme under Grant Agreement No. 945539 Human Brain Project SGA3 and No. 101147319 EBRAINS 2.0 (GTE and TVN). Computational resources for building machine learning models were provided by ACCESS, which is funded by the US National Science Foundation.

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

# A  IBL Brain-wide Map dataset

The full IBL Brain-wide Map dataset contains 699 insertions. Among those, only 675 successfully went through the pre-processing, with 37017 units that pass the IBL spike-sorting quality control. The region distribution of those units are shown in Supplementary Figure 1.

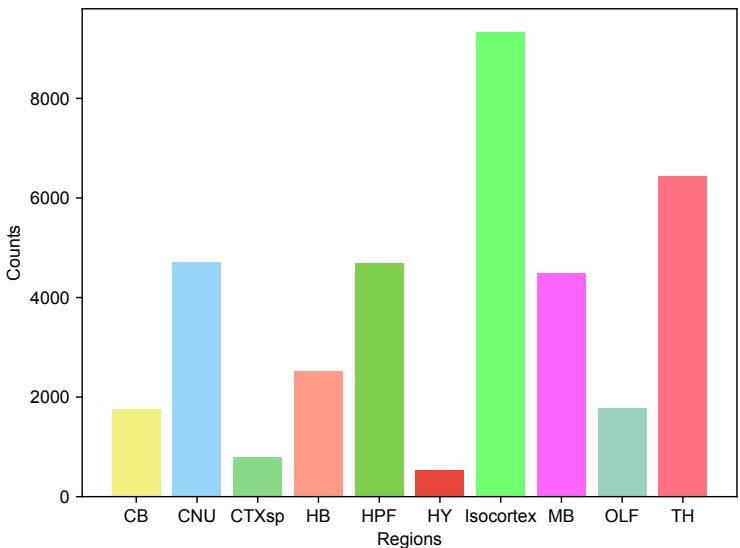

Supplementary Figure 1: Region distribution of the IBL Brain-wide Map dataset. The dataset is very imbalanced and has a small number of units in hypothalamus and cortical subplates.

# B  Baselines and hyperparameters

## B.1  Model Architecture

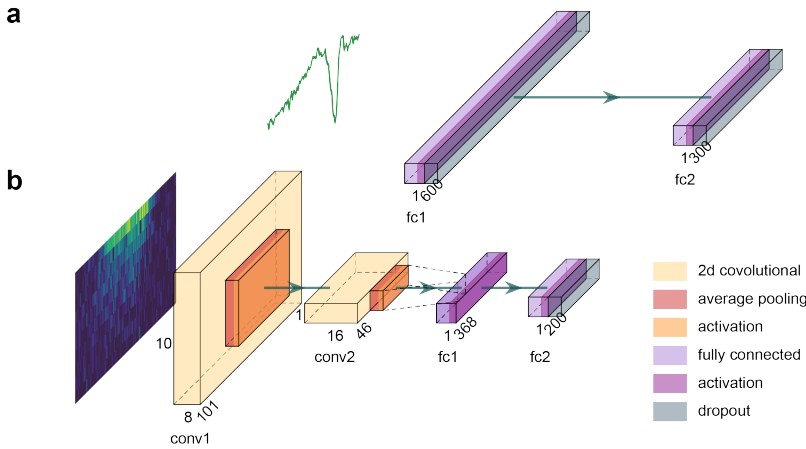

Supplementary Figure 2: Encoder architecture illustration: a) A two-layer multilayer perceptron for waveform encoding, and b) A convolutional neural network for ACG image encoding.

## B.2 MODEL PARAMETERS AND AUGMENTATIONS

We apply the model augmentations in table 1 on the training data during NEMO. $p$ is the probability an augmentation gets applied to an instance. We use the Adam optimizer with learning rate 0.0005 and Cosine Annealing scheduler with $T_0 = 20$. Other model hyperparameters are in table 2. Supplementary Figure 3 shows each of the ACG augmentations on an example ACG image.

Supplementary Table 1: NEMO Augmentations

| Augmentation | Description | Type | p |
|---|---|---|---|
| Gaussian noise | Gaussian noise with mean 0 and std 0.1×std of WVF | WVF | 0.3 |
| Electrode dropout | Randomly zeros out a channel with with a probablity | MC-WVF | 0.1 |
| Amplitude Jitter noise | Rescales a channel's amplitude by a uniform value between 0.9 and 1.1 | MC-WVF | 0.3 |
| Temporal Gaussian smoothing | Smooths an ACG using a Gaussian filter along the temporal axis with $\sigma = 2$ | ACG | 0.5 |
| Temporal jittering | Jitters the temporal axis of an ACG by a random integer between -3 and 3 inclusive | ACG | 0.5 |
| Amplitude scaling | Scales the amplitude of an ACG by a random number between 0.9 and 1.1 | ACG | 0.5 |
| Additive Gaussian noise | Adds Gaussian noise with mean 0 and std 0.1×maximum of ACG | ACG | 0.5 |
| Multiplicative pepper noise | Each value in the ACG has a 5% of being set to 0 | ACG | 0.5 |

Supplementary Table 2: NEMO Hyperparameters

| Parameter | Description | Value |
|---|---|---|
| Epochs | Numbers of epochs to run | 6000 |
| Batch Size | Number of items processed in a single operation | 1024 |
| Learning Rate | Learing rate of the model | 0.0005 |
| Log every n steps | Save model and cross validate results every n epochs | 100 |
| dim_embed | Latent dimension size | 512 |
| Temperature | $\tau$ in the formula for contrastive loss | 0.5 |
| ACG dropout | Dropout probability in ACG encoder | 0.2 |
| Waveform dropout | Dropout probability in waveform encoder | 0.1 |

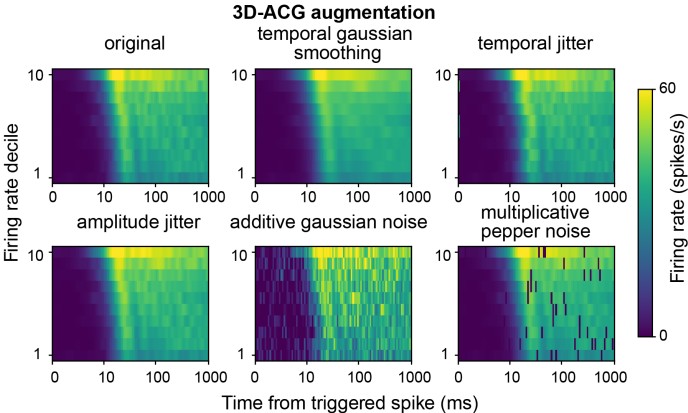

Supplementary Figure 3: Illustration of the augmentations on spiking activity ACGs used by NEMO.

## B.3 MLP FINE-TUNING

For cell-type classification, we fix the hyperparameters of the MLP classifier for all methods to those used in Beau et al. (2025). This is because we have very few labels for evaluation and it is challeng-

ing to hold out any for hyperparameter tuning. For the brain region classification experiments, we tune the optimizer, learning rate, model size and dropout using the validation set of insertions.

### B.4 SimCLR-based NEMO

For the SimCLR-based NEMO, we randomly apply the same set of augmentation methods to each data point, generating two correlated augmented views. We then train each encoder and a linear projection layer to maximize agreement using contrastive learning. Compared to other methods, we use stronger augmentations for SimCLR. This is because SimCLR focuses on the differences between augmentations from the same data point, whereas CLIP-based learning strategies compare augmentations across different modalities. Stronger augmentations improved SimCLR's performance.

Both methods use contrastive learning to maximize similarity between augmented views of the same neuron, while keeping views of different neurons distinct. The main difference between the SimCLR-based method and the CLIP-based method is that CLIP is multimodal and SimCLR is unimodal. For the SimCLR-based method, we use the contrastive objective from (Chen et al., 2020). Let $z_{\text{view1}}$ and $z_{\text{view2}}$ be the normalized projections of two augmented views from the same modality. For a batch $B$, the objective is as follows:

$$
\mathcal{L} = -\frac{1}{2|B|} \sum_{i=1}^{|B|} [\log \frac{\exp(z_{\text{view1}_i} \cdot z_{\text{view2}_i}/\tau)}{\sum_{j=1}^{|B|} \mathbb{1}_{\{j \neq i\}} \exp(z_{\text{view1}_i} \cdot z_{\text{view2}_j}/\tau) + \sum_{j=1}^{|B|} \mathbb{1}_{\{j \neq i\}} \exp(z_{\text{view1}_i} \cdot z_{\text{view1}_j}/\tau)}
$$
$$
+ \log \frac{\exp(z_{\text{view2}_i} \cdot z_{\text{view1}_i}/\tau)}{\sum_{j=1}^{|B|} \mathbb{1}_{\{j \neq i\}} \exp(z_{\text{view2}_i} \cdot z_{\text{view1}_j}/\tau) + \sum_{j=1}^{|B|} \mathbb{1}_{\{j \neq i\}} \exp(z_{\text{view2}_i} \cdot z_{\text{view2}_j}/\tau)}]
\tag{2}
$$

For SimCLR, we use the same types of augmentation as CLIP with increased strength: for the EAPs, we applied Gaussian noise with standard deviation 1 with probability 0.5, rather 0.1 standard with a probability of than a 0.3. For the ACG images, we keep the augmentation methods consistent across all models.

In our SimCLR-based method, we use the pretrained encoder to obtain separate embeddings for each modality, which are then combined to train downstream tasks such as cell-type and brain region classification.

## C  Removing Stimulation artifacts from the NP Ultra data set

The optotagging experiments for the Neuropixels Ultra dataset included two distinct photostimulation patterns: a 10 ms square wave pulse and a 1 s raised cosine ramp, each delivered at one of three light intensities (0.2, 4.1, and 10.0 mW/mm$^2$), randomly interleaved. The average inter-trial interval was 2 s. To eliminate stimulus-induced effects, we excluded spikes occurring within 1100 ms of each stimulus onset and computed ACG images using only the remaining data. The ACG images followed the approach in (Beau et al., 2021), with one modification: we normalized each bin's value solely by the number of spikes whose time lag around that bin's spiking time did not overlap with any stimulus period (i.e., the 1100 ms post-stimulus window). This adjustment mitigates edge effects introduced by blank periods during stimulation.

## D  Dataset split and validation strategy

Given the limited quantity of ground-truth labels for the NP Ultra dataset and the C4 cerebellum dataset, we evaluate the performance of each model using a 5-fold cross-validation of the labeled cells with 10 repeats. Since we do not have a separate validation set while training NEMO, we utilize a nested cross-validation approach to choose an evaluation checkpoint for NEMO. In other words, for each cross-validation fold, we perform a nested 5-fold cross-validation on the *training* folds with a linear classifier to choose the best checkpoint for NEMO. We then use this NEMO checkpoint to perform the evaluation on the heldout fold of the original 5-fold cross-validation. This checkpoint choosing procedure is done only on the training folds and does not use any information from the testing fold.

For the IBL dataset, we use a standard 70-10-20 split for the training, validation, and test sets, respectively. During training, we monitor the performance of the model using a linear classifier

trained on the training set and validated on the validation set. We then compute our evaluation metrics on the test set using the checkpoint with the highest validation F1.

## E   Single-channel NP Ultra cell-type classification

Utilizing single-channel waveforms, NEMO outperforms all baseline methods across all evaluation methods with default parameters. See Supplementary Table 3 for a summary of these results. Compared to the multi-channel results in the main text, however, the performance of all methods are lower, indicating that there is important cell-type information in the multi-channel templates.

Supplementary Table 3: **Cell-type classification for the NP Ultra dataset.** The accuracy and F1-scores are reported for three conditions: (i) a linear layer and (ii) MLP on top of the frozen pretrained representations (for VAE and NEMO), and (iii) after MLP finetuning.

| Model | Linear | | MLP | | MLP fine-tuned | |
|---|---|---|---|---|---|---|
| | Acc | F1 | Acc | F1 | Acc | F1 |
| Supervised | N/A | N/A | N/A | N/A | $0.73 \pm 0.01$ | $0.73 \pm 0.01$ |
| PhysMAP | $0.53 \pm 0.00^2$ | $0.49 \pm 0.00\,^1$ | N/A | N/A | N/A | N/A |
| VAE (10d latent) | $0.74 \pm 0.01$ | $0.74 \pm 0.01$ | $0.74 \pm 0.00$ | $0.74 \pm 0.00$ | $0.75 \pm 0.00$ | $0.75 \pm 0.00$ |
| VAE (500d rep) | $0.74 \pm 0.01$ | $0.74 \pm 0.01$ | $0.74 \pm 0.01$ | $0.75 \pm 0.01$ | $0.76 \pm 0.01$ | $0.76 \pm 0.00$ |
| NEMO (500d rep) | **$0.75 \pm 0.01$** | **$0.75 \pm 0.01$** | **$0.77 \pm 0.01$** | **$0.77 \pm 0.01$** | **$0.78 \pm 0.00$** | **$0.78 \pm 0.00$** |

## F   Cell-type classification on the C4 cerebellum data

We evaluate NEMO in comparison to all baseline models on the C4 cerebellum opto-tagged dataset using 5-fold cross validation. We utilize the same training pipeline for the VAE as proposed in Beau et al. (2025) including the data augmentations. Supplementary Figure 4 show the results of this analysis. Across all evaluation schemes, NEMO outperforms the baseline models.

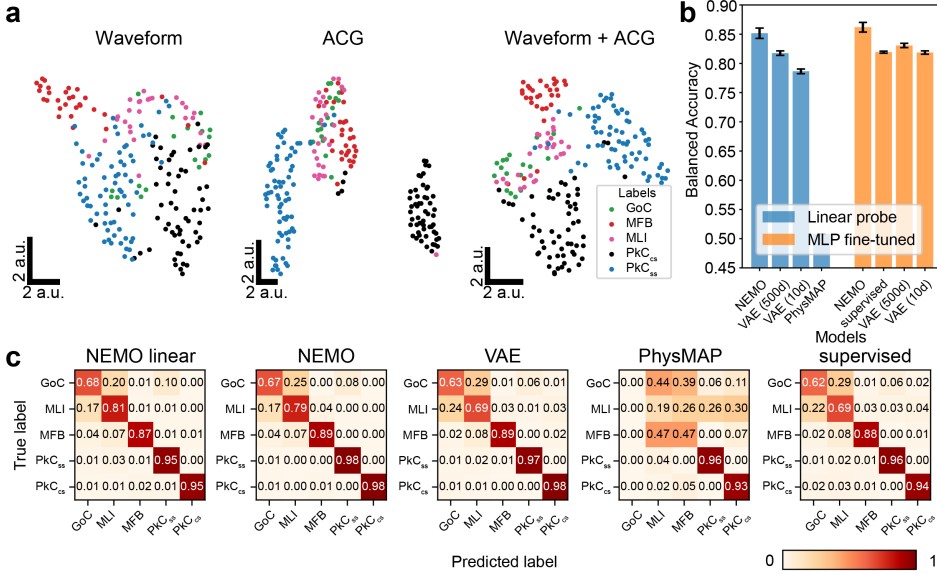

Supplementary Figure 4: **Comparing NEMO to baseline models on the C4 cerebellum opto-tagged dataset.** (a) UMAP visualization of the pretrained NEMO representations of unseen opto-tagged cerebellum units, colored by different cell-types. Neurons of the same class form clusters, particularly when combined modalities are used. (b) Balanced accuracy and (c) Confusion matrices for the C4 cerebellum classification results, normalized by ground truth label and averaged across 5 random seeds. NEMO outperforms the other embedding methods by a significant margin across all cell-types and evaluation methods.

## G  TRANSFER LEARNING BY PRETRAINING ON THE IBL BRAIN-WIDE MAP

After pretraining NEMO on the IBL Brain-wide Map dataset, we evaluated how well the model performs at single-channel cell-type classification of the NP Ultra dataset before and after fine-tuning. For this experiment, we have three models: (1) NEMO (Ultra) pretrained and evaluated on NP Ultra data, (2) NEMO (IBL) pretrained on IBL data and evaluated on NP Ultra data, (3) NEMO (IBL) pretrained on IBL data and NP Ultra data and evaluated on NP Ultra data. Interestingly, we found that the pretrained IBL model performed well on cell-type classification even without training examples from NP Ultra (see Supplementary Table 4).

Supplementary Table 4: **Transfer learning using IBL data for pretraining and the NP Ultra dataset for evaluation.** The accuracy and F1-scores are reported for three conditions: (i) a linear layer and (ii) MLP on top of the frozen pretrained representations (for VAE and NEMO), and (iii) after MLP finetuning.

| Model | Linear | | MLP | | MLP fine-tuned | |
|---|---|---|---|---|---|---|
| | Acc | F1 | Acc | F1 | Acc | F1 |
| NEMO (Ultra) | $0.75 \pm 0.01$ | $0.75 \pm 0.01$ | $\mathbf{0.77 \pm 0.01}$ | $\mathbf{0.77 \pm 0.01}$ | $\mathbf{0.78 \pm 0.00}$ | $\mathbf{0.78 \pm 0.00}$ |
| NEMO (IBL) | $0.75 \pm 0.01$ | $0.75 \pm 0.01$ | $\mathbf{0.77 \pm 0.01}$ | $\mathbf{0.77 \pm 0.01}$ | $\mathbf{0.78 \pm 0.00}$ | $\mathbf{0.78 \pm 0.00}$ |
| NEMO (IBL+Ultra) | $\mathbf{0.76 \pm 0.01}$ | $\mathbf{0.76 \pm 0.01}$ | $\mathbf{0.77 \pm 0.01}$ | $\mathbf{0.77 \pm 0.01}$ | $0.77 \pm 0.01$ | $0.78 \pm 0.01$ |

## H  MULTI-CHANNEL IBL BRAIN REGION CLASSIFICATION

For the IBL multichannel experiments, we use 25 channel templates centered on the peak channel if possible. We add two additional template augmentations: amplitude jitter (Vishnubhotla et al., 2024) and electrode dropout. Amplitude jitter rescales a channel's amplitude by a uniform value between 0.9 and 1.1 with $p = 0.3$. Electrode dropout zeros out a channel with $p = 0.1$. If all channels are zeroed out, we leave the peak channel. Our model architecture is the same as NEMO with the only difference being the input size of the waveform encoder.

We tune hyperparameters for both the linear and MLP classification models. For the linear model, we tune the inverse of the regularization strength between $1e - 5$ and $1e4$ using the python module optuna (Akiba et al., 2019). For the MLP model, we do a grid search over the dropout probability$(0.1, 0.2, 0.3, 0.4)$, learning rate$(1e - 4, 1e - 5, 1e - 6)$, number of layers$(1, 2, 3)$ and layer size$(128, 256, 512)$.

Supplementary Table 5: Multi-channel, single-neuron brain region decoding for the IBL dataset

| Model | Linear | | MLP | | MLP fine-tuned | |
|---|---|---|---|---|---|---|
| | Acc | F1 | Acc | F1 | Acc | F1 |
| NEMO | $0.42 \pm 0.00$ | $0.40 \pm 0.00$ | $0.45 \pm 0.01$ | $0.42 \pm 0.00$ | $0.47 \pm 0.01$ | $0.44 \pm 0.01$ |
| NEMO (25-channel) | $\mathbf{0.45 \pm 0.01}$ | $\mathbf{0.42 \pm 0.00}$ | $\mathbf{0.46 \pm 0.00}$ | $\mathbf{0.43 \pm 0.00}$ | $\mathbf{0.48 \pm 0.00}$ | $\mathbf{0.45 \pm 0.00}$ |

Supplementary Table 6: multi-neuron, multi-channel brain region classification for the IBL dataset

| Model | Linear | | MLP | | MLP fine-tuned | |
|---|---|---|---|---|---|---|
| | Acc | F1 | Acc | F1 | Acc | F1 |
| NEMO | $0.48 \pm 0.00$ | $0.45 \pm 0.00$ | $0.50 \pm 0.00$ | $\mathbf{0.48 \pm 0.00}$ | $0.51 \pm 0.00$ | $0.50 \pm 0.00$ |
| NEMO (25-channel) | $\mathbf{0.50 \pm 0.01}$ | $\mathbf{0.47 \pm 0.00}$ | $\mathbf{0.51 \pm 0.01}$ | $\mathbf{0.48 \pm 0.01}$ | $\mathbf{0.52 \pm 0.00}$ | $\mathbf{0.51 \pm 0.01}$ |

## I  PICKING PARAMETERS FOR IBL UNIT REPRESENTATION CLUSTERING

We used the Python-implemented UMAP package (McInnes et al., 2018b) and the Python-Louvain package (Aynaud, 2020) for our clustering analyses. For our clustering analysis, we aim to find the most informative clustering with the smallest number of clusters. There are two parameters to tune: 1) the size of local neighborhood used for the UMAP graph manifold approximation resolution that controls the community size in Louvain clustering ($n_{neighbor}$), and 2) the resolution $\gamma$ that

determines the size of the communities. We tuned $n_{neighbor}$ while keeping the resolution to be 1.0 and tracked the final number of clusters. We picked the 'elbow' that has the smallest $n_{neighbor}$ (200). The resolution was chosen by maximizing the modularity index $Q$ of Louvain clustering with $n_{neighbor} = 200$. The modularity index of a graph is defined as:

$$Q = \sum_{c=1}^{n} \left[ \frac{L_c}{m} - \gamma \left( \frac{k_c}{2m} \right)^2 \right]$$

where each $c$ represents a community in the graph, $m$ is the number of edges, $L_c$ is the number of intra-community links for community c, $k_c$ is the sum of degrees of the nodes in community $c$, and $\gamma$ is the resolution parameter.

## J  PARAMETERS FOR CLASSIFIERS AND FINE-TUNING METHODS

Supplementary Table 7: Linear probe best hyperparameters

| Hyperparameter | Value |
|---|---|
| $max_{iter}$ | 1000 |
| $tol$ | 1e-5 |
| NP Ultra Celltype $c$ | 0.02 |
| IBL NEMO joint $c$ | 0.02 |
| IBL VAE $c$ | 0.4 |
| IBL supervise $c$ | 0.001 |
| IBL NEMO independent $c$ | 2.5 |

Supplementary Table 8: IBL MLP hyperparameters

| Hyperparameter | Value |
|---|---|
| $n_{layer}$ | 1 |
| layer0 size | 256 |
| Dropout rate | 0.2 |
| Supervised scheduler | CosineAnnealingWarmRestarts |
| Supervised $T_0$ | 20 |
| Supervised $T_{mult}$ | 1 |
| Other models' schedular | StepLR |
| StepLR $step_{size}$ | 200 |
| StepLR $\gamma$ | 0.8 |
| Supervised lr | $1 \times 10^{-4}$ |
| Other models' lr | $1 \times 10^{-5}$ |

## J.1 UMAP EMBEDDINGS WITH RAW FEATURES, RANDOM PROJECTION AND VAE

We visualize the UMAP embeddings of the raw features, randomly initialized encoder projection and the VAE projection for the C4 cerebellum dataset in Supplementary figure 5. We find that NEMO representations (shown in Supplementary Figure 4) are visually more structured.

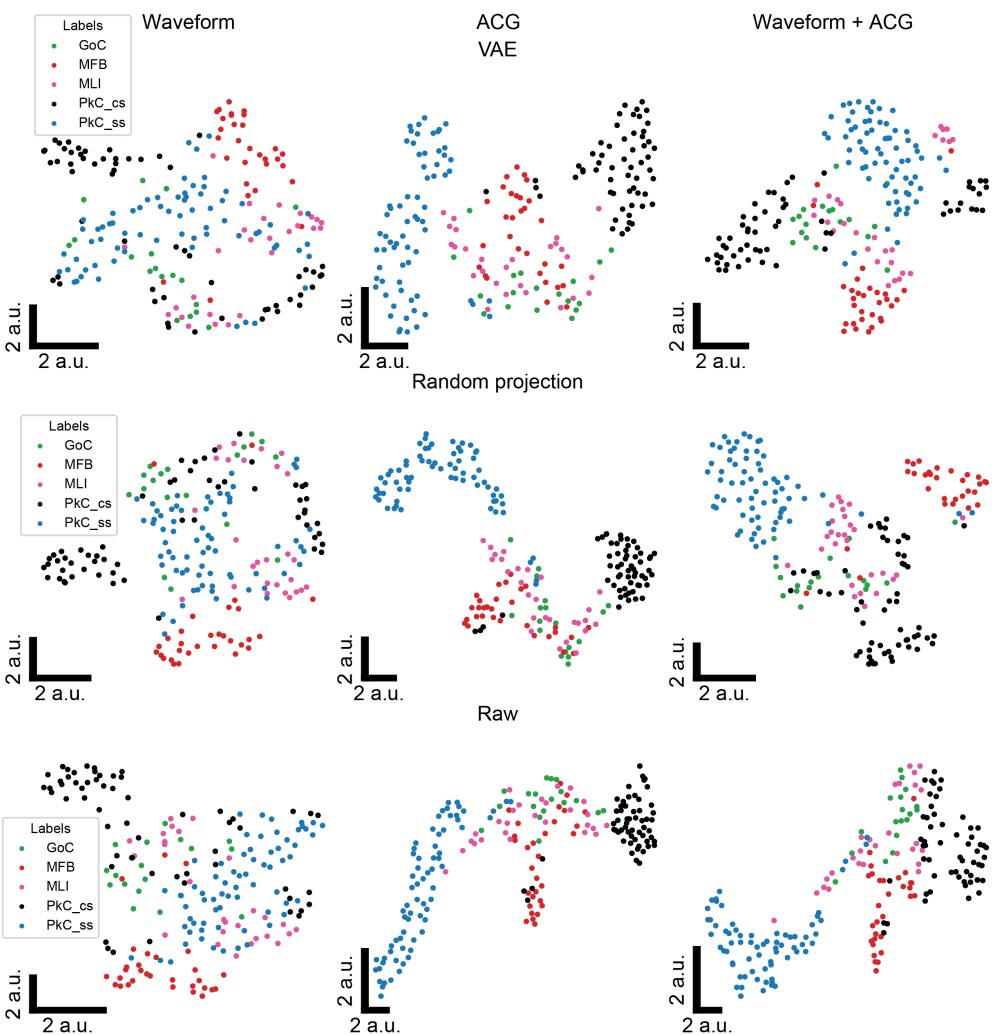

Supplementary Figure 5: UMAP visualizations using VAE, random projections, and raw features. The random projections are representations which are passed through randomly initialized encoders with the same architecture as NEMO.

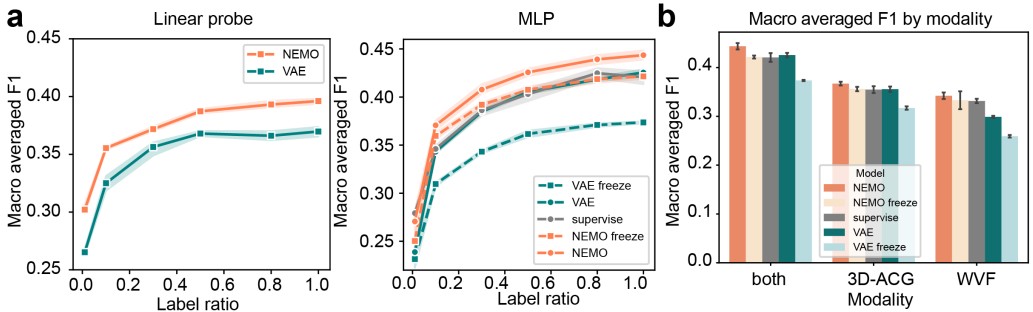

Supplementary Figure 6: Left and middle, macro-averaged F1 scores for linear and MLP-based single-neuron classification of brain region by label ratio, as in Fig. 3(d), replacing accuracy with F1 score. Right, single-neuron MLP-classification balanced F1 scores for uni- and bimodal models, as in Fig. 3(e), replacing accuracy with F1 score.

## J.2 IBL REGION CLASSIFICATION WITH LABEL RATIO SWEEP

In Supplementary Figure 6, we show the macro-averaged F1 scores for single neuron classification of brain region by label ratio as complementary to Figure 3.

In Supplementary Figure 7, we study the effect of varying the ratio of labeled data used to train the brain region classifier. Fully supervised methods used the same labeled examples, but were not pretrained. Means and standard-deviation bands are computed over five random initializations for each label ratio. Supplementary Figure 8 shows the macro-averaged F1 scores for single neuron classification of brain region by label ratio with the IBL dataset, complementary to Figure 5.

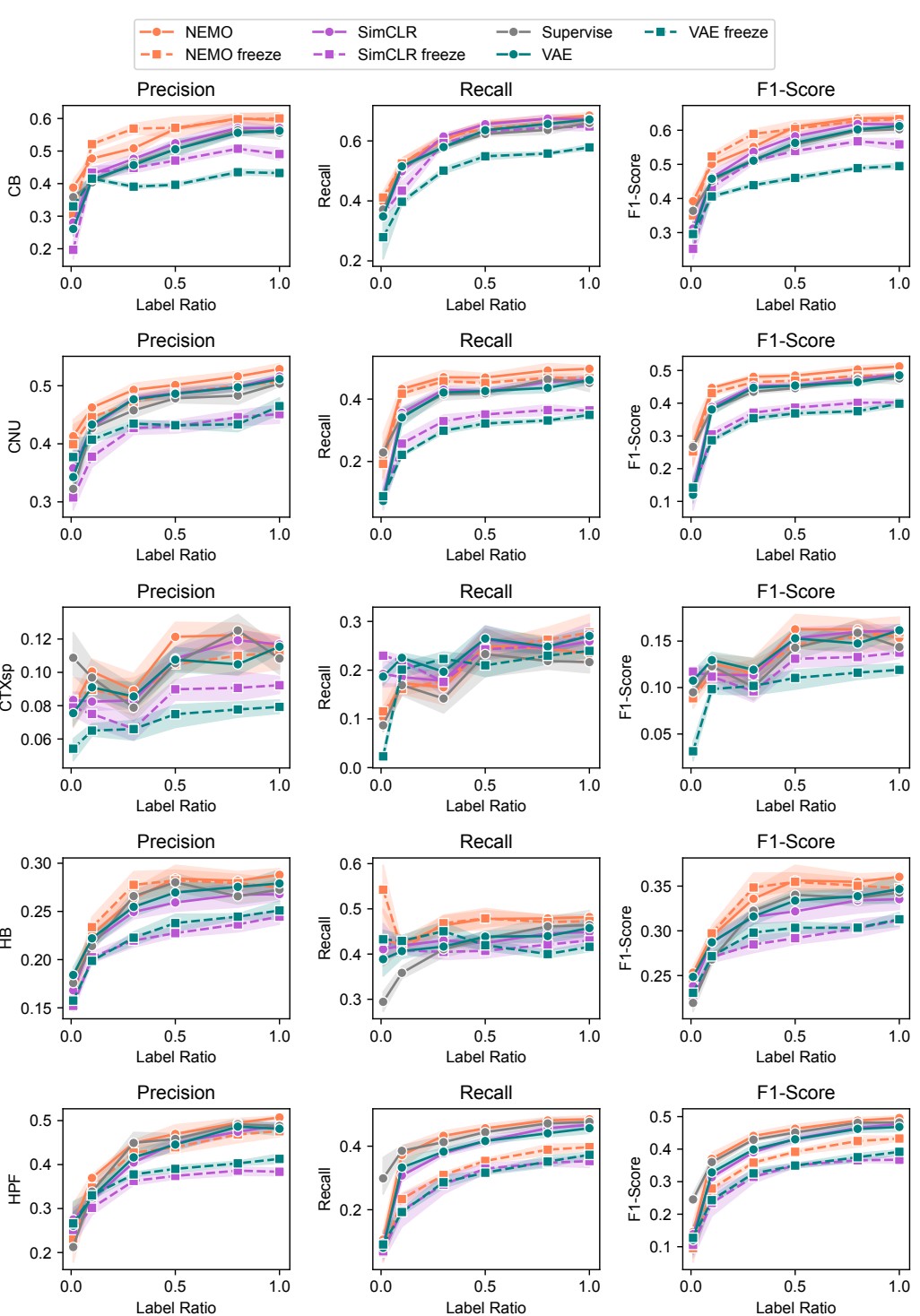

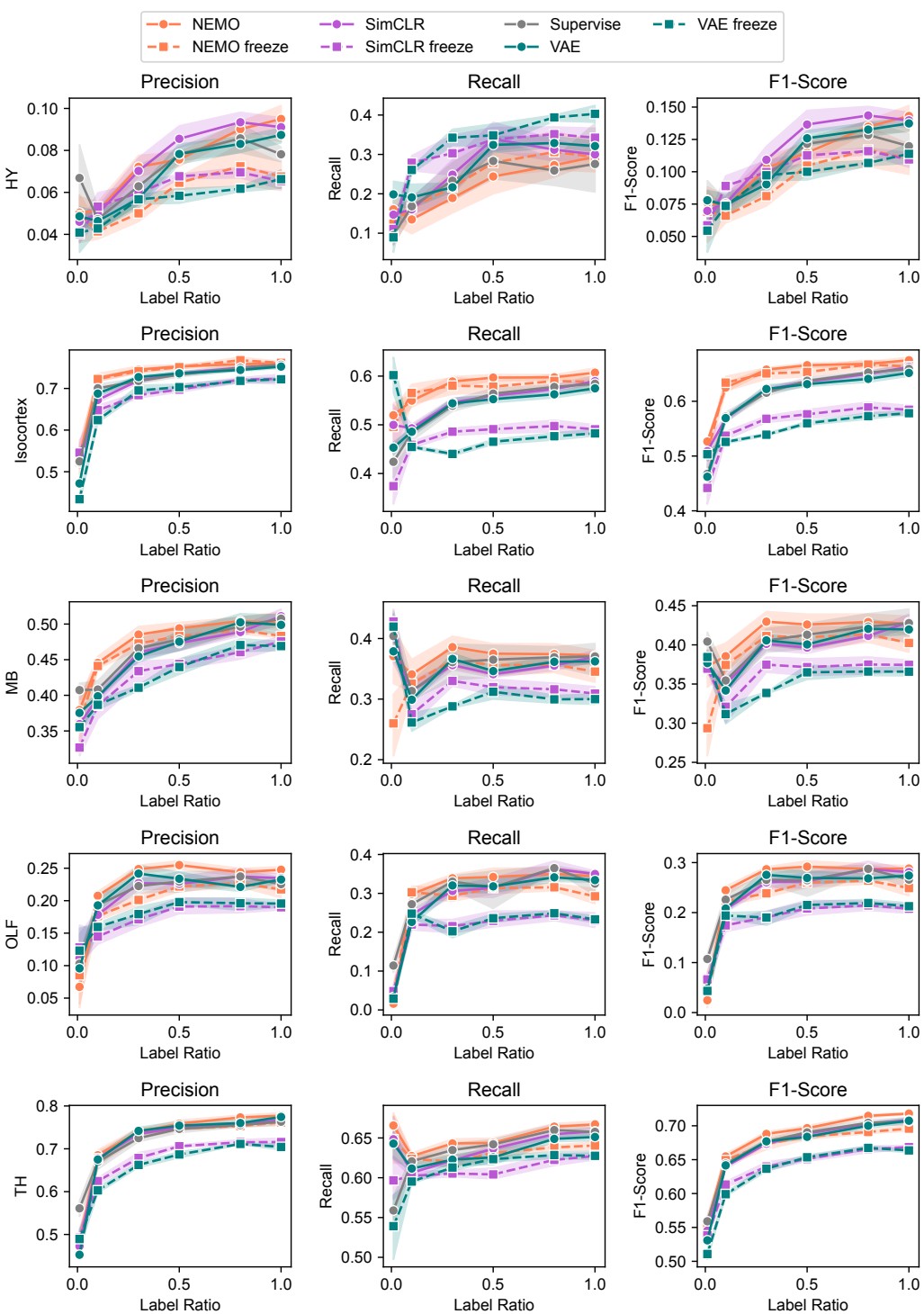

Supplementary Figure 7: Detailed comparison of NEMO and SimCLR for region classification performance on IBL data with varying label ratio. We show the precision, recall and F1 score for each class. NEMO shows superior performance in most of the classes, except hypothalamus (HY), which has a small sample size compared to other classes.

## K  INDEPENDENT LEARNING ABLATION FOR NEMO

We ablate the cell-type classification performance between NEMO with independent (SimCLR) and joint learning (CLIP). The results are shown in Supplementary table 9 and Supplementary table 10. Joint training improves cell-type classification for both datasets.

Supplementary Table 9: Single-channel cell-type classification for the NP Ultra dataset for independent vs. joint learning NEMO.

| Model | Linear | | MLP | | Finetuned MLP | |
|---|---|---|---|---|---|---|
| | Acc | F1 | Acc | F1 | Acc | F1 |
| independent NEMO | $0.74 \pm 0.05$ | $0.74 \pm 0.01$ | $0.75 \pm 0.01$ | $0.75 \pm 0.00$ | $0.76 \pm 0.02$ | $0.77 \pm 0.00$ |
| joint NEMO | $\mathbf{0.75 \pm 0.01}$ | $\mathbf{0.75 \pm 0.01}$ | $\mathbf{0.77 \pm 0.01}$ | $\mathbf{0.77 \pm 0.01}$ | $\mathbf{0.78 \pm 0.00}$ | $\mathbf{0.78 \pm 0.00}$ |

Supplementary Table 10: Single-channel cell-type classification for the C4 dataset for independent vs. joint learning NEMO

| Model | Linear | | MLP (5-fold) | | Finetuned MLP (5-fold) | |
|---|---|---|---|---|---|---|
| | Acc | F1 | Acc | F1 | Acc | F1 |
| independent NEMO (500d rep) | $0.82 \pm 0.00$ | $0.81 \pm 0.00$ | $0.82 \pm 0.01$ | $0.82 \pm 0.00$ | $0.84 \pm 0.01$ | $0.84 \pm 0.01$ |
| joint NEMO (500d rep) | $\mathbf{0.85 \pm 0.01}$ | $\mathbf{0.85 \pm 0.01}$ | $\mathbf{0.85 \pm 0.00}$ | $\mathbf{0.85 \pm 0.00}$ | $\mathbf{0.86 \pm 0.01}$ | $\mathbf{0.86 \pm 0.01}$ |

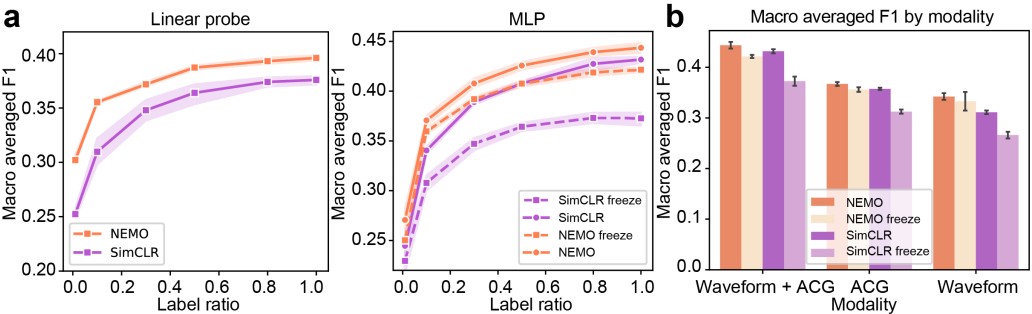

Supplementary Figure 8: Left and middle, macro-averaged F1 scores for linear and MLP-based single-neuron classification of brain region by label ratio for independent learning ablation, as in Fig. 5(a), replacing accuracy with F1 score. Right, single-neuron MLP-classification balanced F1 scores for uni- and bimodal models, as in Fig. 5(b), replacing accuracy with F1 score.

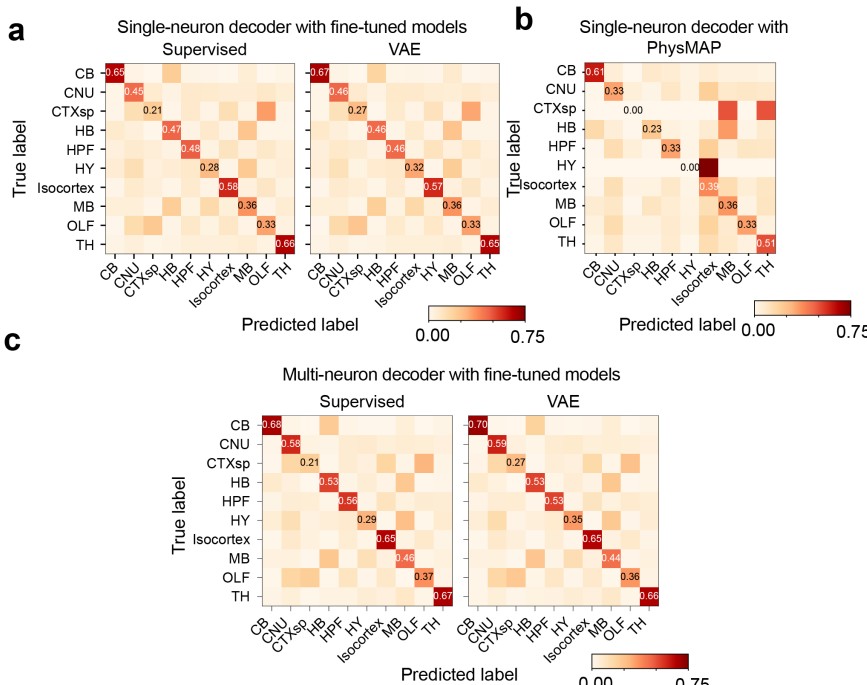

Supplementary Figure 9: Classification results for the VAE, supervised MLP, and PhysMAP on the IBL brain region classification task. Due to label imbalance, PhysMAP is unable to predict CTXsp and HY, which leads to a low balanced accuracy and F1 score.

## L IBL UNIT CLUSTERING RESULTS WITH NEMO

Supplementary Figure 10 shows averages and standard deviations for each cluster's template waveforms and ACG images. Supplementary Figure 11 shows the distribution of cluster labels over brain regions, separated by individual insertions and by the recording lab.

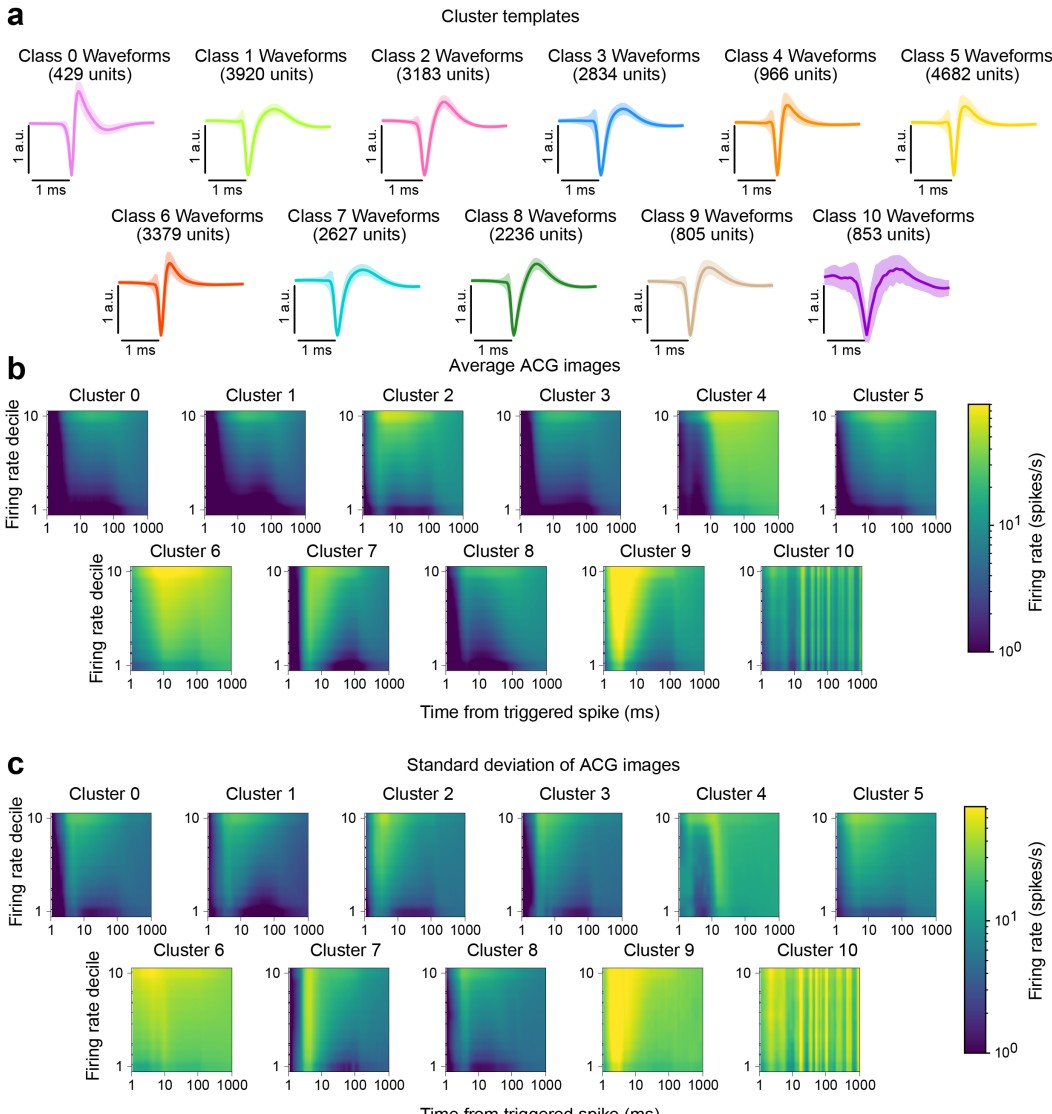

Supplementary Figure 10: Average and standard deviation of the template waveforms and ACGs for the units in each cluster, as shown in Figure 4. (a) The waveforms are consistent within clusters and distinct across clusters. (b) The ACG images are are also distinct across clusters. These results suggest that NEMO is able to find distinct clusterings of neurons across the whole-brain.

## M IBL UNIT CLUSTERING RESULTS WITH RAW FEATURE INPUT

Supplementary Figure 12 shows clustering results for the IBL dataset. The hyper-parameters were selected with similar criteria as in Section 6. Since the number of clusters does not show a similar 'elbow,' but keeps decreasing as $n_{neighbor}$ increases, we picked $n_{neighbor} = 1000$ and used a resolution $\gamma$ that maximizes the the modularity and minimizes the number of clusters. These clusters are less spatially organized compared to the clusters clustered using NEMO. As shown in Supplemen-

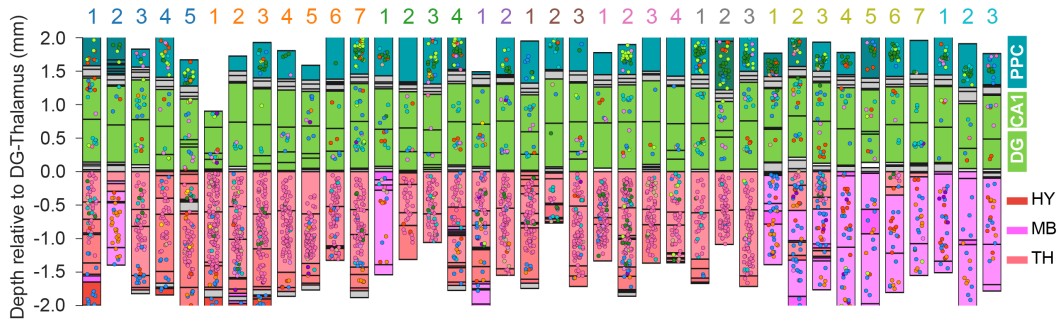

Supplementary Figure 11: Visualization of clustering results across repeated site in the IBL dataset (IBL et al., 2022). The Neuropixels probes target the same brain regions (including posterior parietal cortex, hippocampus, and thalamus) in these insertions. The color of the labels on top of each column indicates the lab ID of each insertion. Our results reveal that the clusters are highly distinguishable by region, with each region containing a distinct group of neurons. Moreover, the dominant cluster IDs for the same region remain consistent across different insertions.

tary Figure 13, the clustering result based on NEMO overall shows lower entropy, which indicates higher region-selectivity.

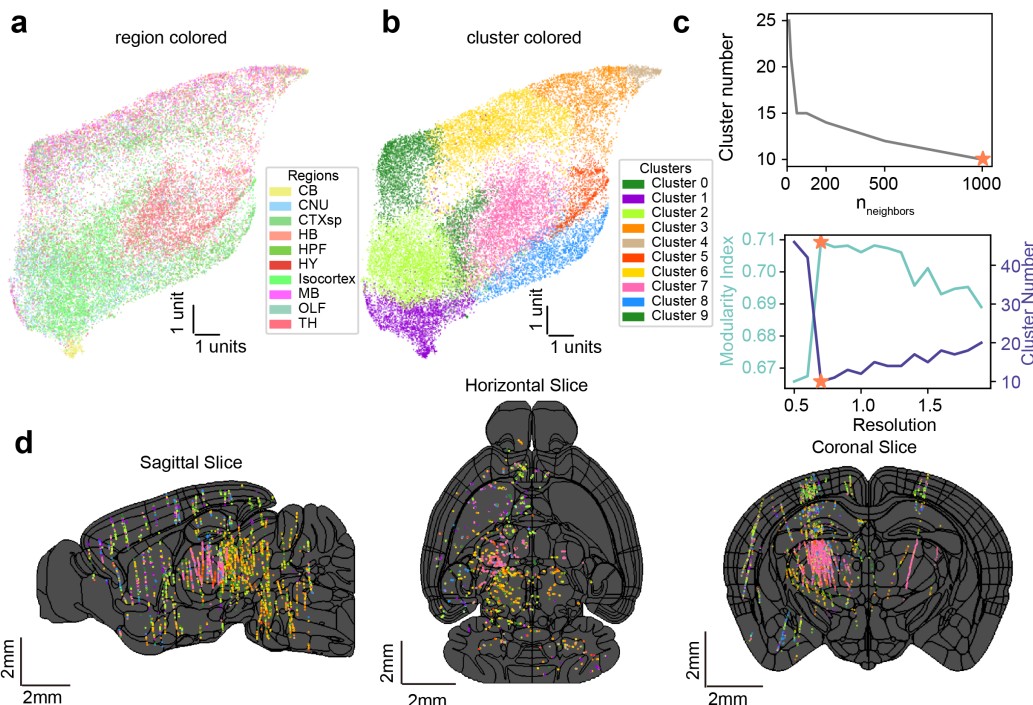

Supplementary Figure 12: IBL neuron clustering using raw features. (a) A UMAP visualization of raw features on the training data colored by anatomical brain region. (b) The same UMAP as shown in (a) but instead colored by cluster labels using a graph-based approach (Louvain clustering). (c) We tuned the neighborhood size in UMAP and the resolution for the clustering. These parameters were selected by maximizing the modularity index which minimized the number of clusters. (d) 2D brain slices across three brain views with the location of individual neurons colored using the cluster IDs shown in (b). The black lines show the region boundaries of the Allen mouse atlas.

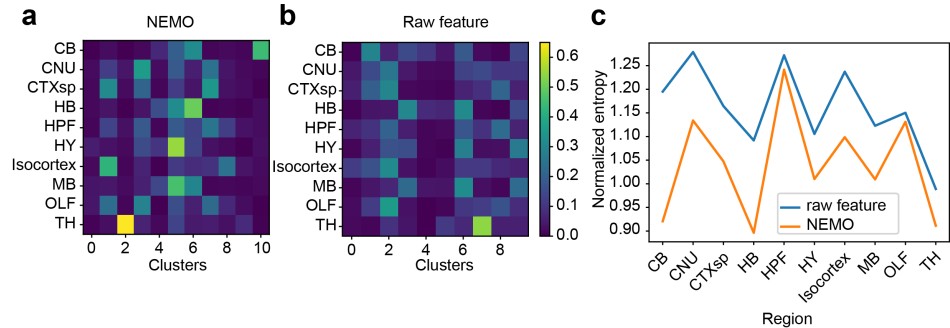

Supplementary Figure 13: Region distribution of clustering results using different NEMO features and raw features (normalized by region). For each region, we get a cluster distribution vector. We then computed the normalized entropy of that distribution. Cluster result based on NEMO overall shows lower entropy, which indicates higher region-selectivity.

# N SENSITIVITY TO PARAMETERS

## N.1 MULTI-CHANNEL NP ULTRA HYPERPARAMETER SWEEP

To assess the sensitivity of hyperparameters for NEMO and the VAE baseline, we specify a range of values for each hyperparameter and randomly sample across 50 models on one seed. For the learning rate, we draw samples from a log-uniform distribution between 1e-4 and 5e-3. Dropout values are sampled from a uniform distribution between 0 and 0.4. The latent dimension size is randomly selected from the set (10, 20, 256, 512, 1024). For the VAE, the learning rate is restricted to the range 1e-4 to 5e-4 to prevent gradient explosions. We report both the median and best performance for each model across hyperparameter configurations. Notably, since this dataset lacks a validation set, the best-performing model is reported for illustrative purposes only. Supplementary Table 11 shows the median performance and Supplementary Table 12 shows the best performance. We find that NEMO outperforms the VAE-based model for both the median and best performing models.

Supplementary Table 11: **Median performance of the VAE and NEMO model for different hyperparameters.** The accuracy and F1-scores are reported for three conditions: (i) a linear layer and (ii) MLP on top of the frozen pretrained representations (for VAE and NEMO), and (iii) after MLP finetuning.

| Model | Linear | | MLP | | MLP fine-tuned | |
|---|---|---|---|---|---|---|
| | Acc | F1 | Acc | F1 | Acc | F1 |
| VAE (latent; Beau et al. (2025)) | 0.74 | 0.73 | 0.73 | 0.72 | 0.78 | 0.78 |
| VAE (representation) | 0.76 | 0.76 | 0.77 | 0.77 | 0.78 | 0.79 |
| NEMO | **0.78** | **0.78** | **0.80** | **0.79** | **0.80** | **0.80** |

Supplementary Table 12: **Best evaluation performance of the VAE and NEMO model for different hyperparameters.** The accuracy and F1-scores are reported for three conditions: (i) a linear layer and (ii) MLP on top of the frozen pretrained representations (for VAE and NEMO), and (iii) after MLP finetuning.

| Model | Linear | | MLP | | MLP fine-tuned | |
|---|---|---|---|---|---|---|
| | Acc | F1 | Acc | F1 | Acc | F1 |
| VAE (latent; Beau et al. (2025)) | 0.77 | 0.76 | 0.77 | 0.77 | 0.79 | 0.79 |
| VAE (representation) | 0.77 | 0.77 | 0.79 | 0.79 | 0.80 | 0.80 |
| NEMO | **0.80** | **0.80** | **0.81** | **0.81** | **0.81** | **0.81** |

## N.2 AUGMENTATION ABLATIONS

To assess the importance of augmentations, we conducted an ablation experiment in two parts: (1) starting with the full set of augmentations, we removed one augmentation at a time, and (2) starting with no augmentations, we added one augmentation at a time. The resulting balanced accuracy and macro-averaged F1 scores are presented in Supplementary Figure 14. The results indicate that additive Gaussian noise in the ACG images is the most impactful augmentation, while the absence of augmentations significantly degrades performance. For other augmentations, the combined effect is large while the individual contribution is smaller.

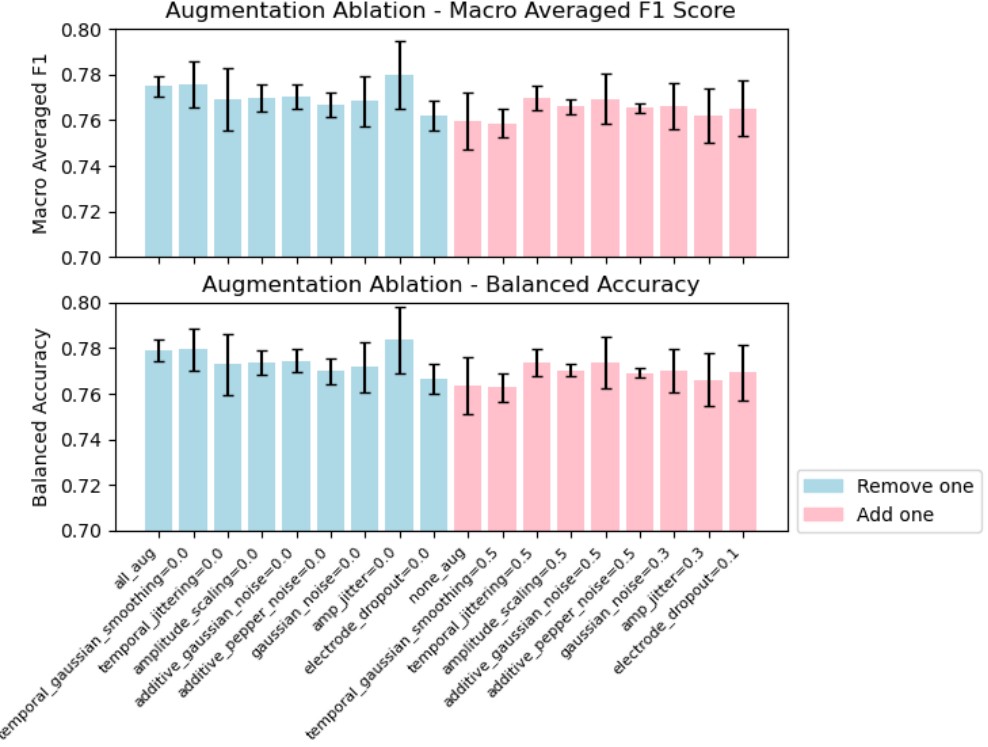

Supplementary Figure 14: To assess the impact of our augmentations, we performed two analyses on the multi-channel UHD cell-type classification dataset: (1) removing one augmentation at a time, starting from the full set of augmentations, and (2) applying one augmentation at a time, starting from no augmentations. For each condition, we calculated the mean and standard deviation of (a) balanced accuracy and (b) macro-averaged F1 score with linear classifier.

### N.3 MULTI-NEURON NEIGHBOR SWEEP

To evaluate the significance of the number of neurons we choose in our multi-neuron brain region classification model, we conduct a sweep over the maximum number of neighboring neurons (3, 5, 9, 15, 25) that can be selected within a 60-micron radius on the IBL dataset. As illustrated in Supplementary Figure 15, the overall classification performance improves as the maximum number of neighboring neurons increases, reaching a saturation point when this number reaches $\sim$10.

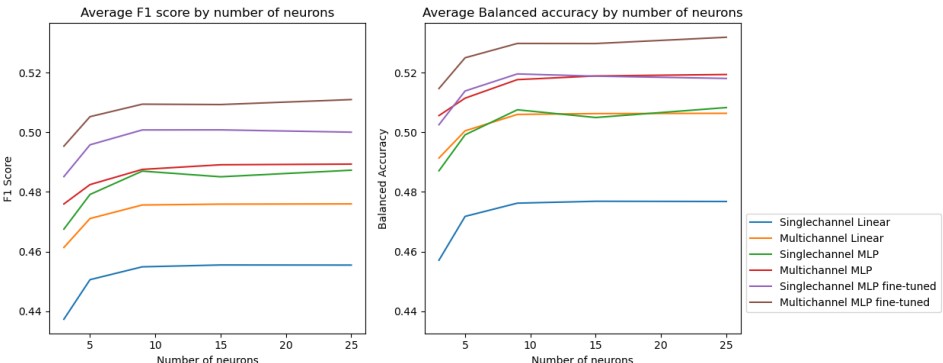

Supplementary Figure 15: Macro-averaged F1 score and Balanced accuracy over 5 seeds of NEMO with multineuron classifier over different number of max neighboring neurons for single and multi-channel IBL with linear, frozen MLP, and MLP fine-tuning.

### N.4 IBL CLUSTERING WITH DIFFERENT RANDOM SEEDS

We trained five instances of NEMO with different random seeds and clustered the neurons using the same hyperparameters. The clustering results are visualized in Supplementary Figure 16, where UMAP was applied for dimensionality reduction, with colors distinguishing clusters. To ensure direct comparison, we maintain consistent UMAP visualizations across clustering results. The average adjusted Rand Index (ARI) for these outcomes is $0.48 \pm 0.04$, while the adjusted mutual information (AMI) score is $0.58 \pm 0.02$. These metrics indicate a moderate level of agreement among the clustering results.

The boundaries of the clusters are subtle, introducing stochasticity into the clustering results. In addition, clustering the full brain dataset results in a coarse segmentation, while significant variability is expected within individual regions. Therefore, these clustering results should mainly be interpreted as exploratory.

## O STATISTICAL TESTS

To evaluate performance differences between models, we conducted two-sample one-tailed t-tests (significance threshold $p < .05$) on the macro-averaged F1 scores and balanced accuracies across five random seeds. The results are summarized in a significance matrix comparing models: red indicates the model in the column is significantly better than the model in the row, blue indicates it is significantly worse, and white indicates no significant difference.

### O.1 CELL-TYPE CLASSIFICATION

**Multi-channel NP Ultra opto-tagged mouse data** For both linear and MLP classifiers, NEMO significantly outperforms all baselines (Supplementary Figure 17).

**C4 opto-tagged mouse data.** For both linear and MLP classifiers, NEMO significantly outperforms all baselines (Supplementary Figure 18).

### O.2 BRAIN REGION CLASSIFICATION

NEMO consistently outperforms all other baselines for both linear and MLP classifiers (Supplementary Figure 19). For single-modality brain region classification with the MLP classifier (Sup-

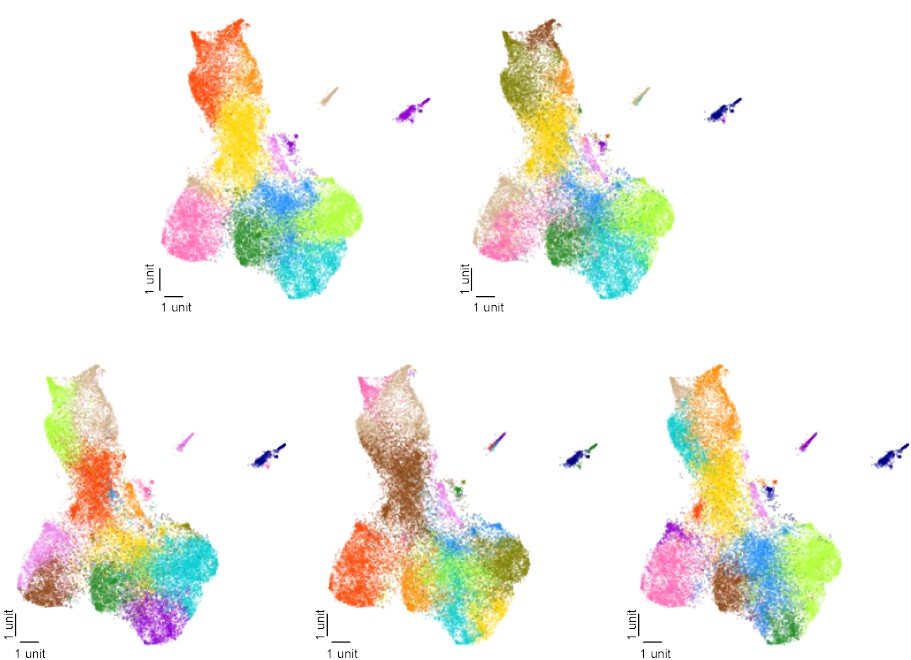

Supplementary Figure 16: Clustering results from five NEMO instances trained with different random seeds, visualized using UMAP with consistent dimensionality reduction across runs. Colors indicate distinct clusters.

plementary Figure 20), NEMO with the fine-tuned MLP either significantly outperforms or shows no significant difference compared to all other baselines.

## O.3 COMPARISON ACROSS LABEL RATIOS

We further compared classifier performance under varying label ratios. For both the linear classifier (Supplementary Figures 21 and 22) and MLP classifier (Supplementary Figures 23 and 24), NEMO significantly outperforms other baselines at each label ratio. Notably, with only 50% of the labels, the linear classifier with NEMO either significantly outperforms or shows no significant difference compared to all baselines using 100% of the labels. Similarly, the MLP classifier with NEMO achieves comparable or significantly better performance than all baselines using 100% of the labels with only 80% of the labels.

Overall, these results demonstrate that NEMO consistently outperforms or matches the performance of other baselines across multiple evaluation criteria, including cell-type classification and brain region decoding tasks. Its robustness is evident across classifiers, label ratios, and metrics, establishing NEMO as a highly effective model for these challenging neurobiological classification tasks.

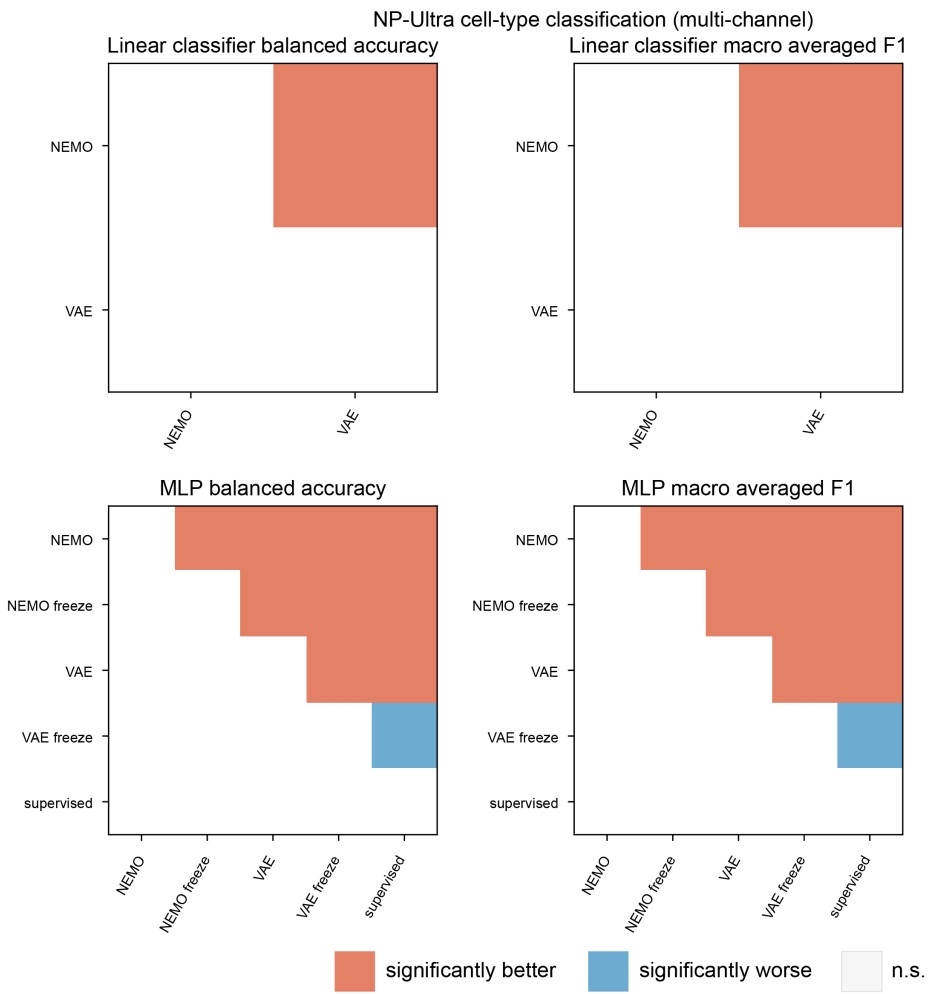

Supplementary Figure 17: One-tailed t-test results for NP Ultra cell-type classification. For both linear and MLP classifiers, NEMO significantly outperforms or matches all baselines.

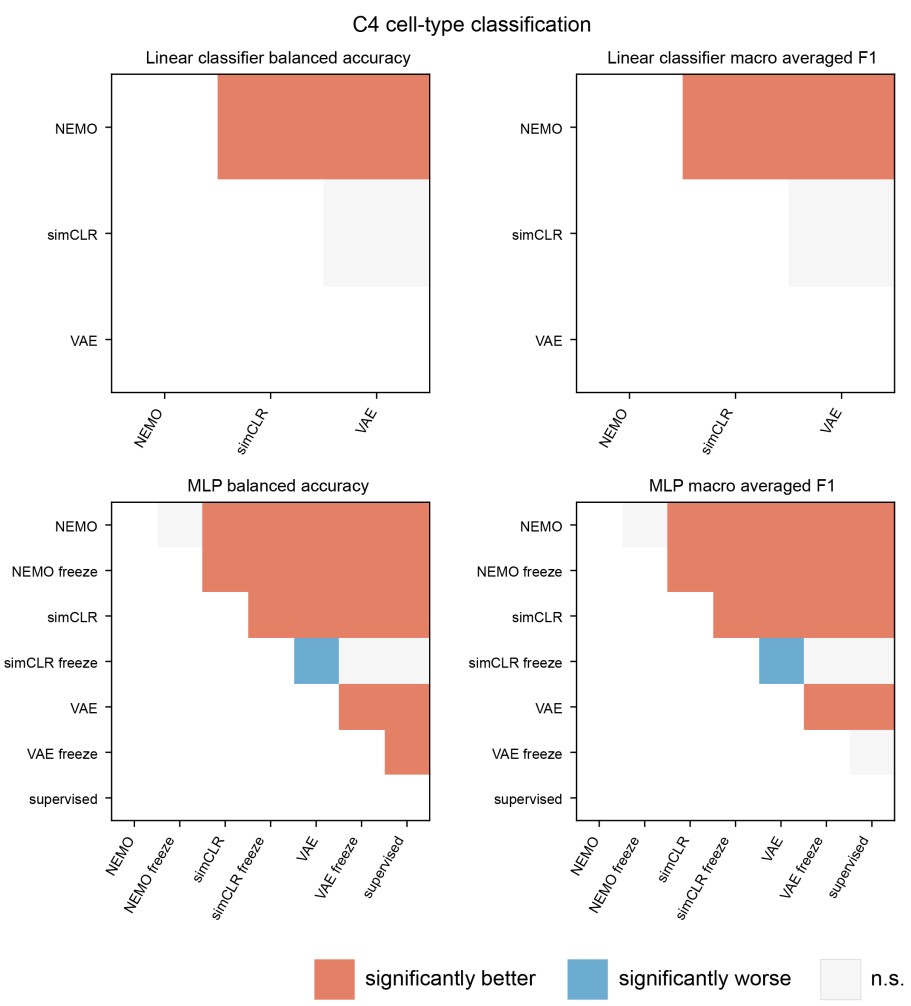

Supplementary Figure 18: One-tailed t-test results for NP Ultra cell-type classification. For both linear and MLP classifiers, NEMO significantly outperforms or matches all baselines.

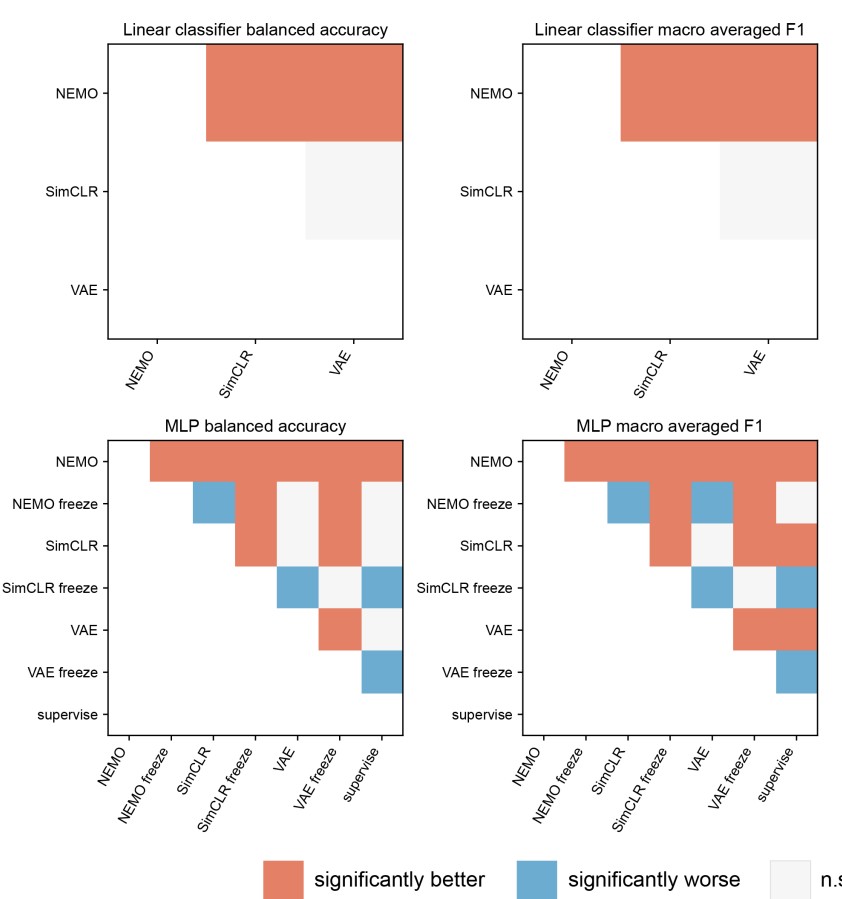

Supplementary Figure 19: One-tailed t-test results for IBL brain region classification. For both linear and MLP classifiers, NEMO significantly outperforms all baselines.

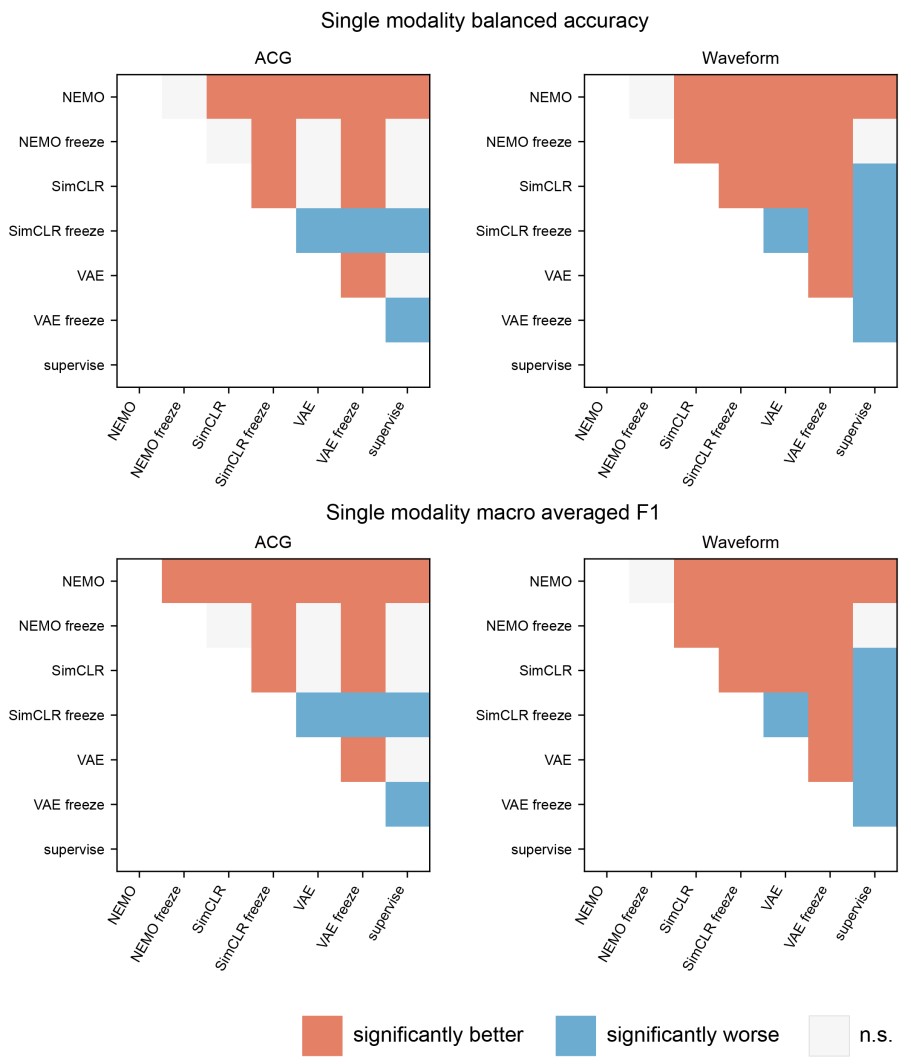

Supplementary Figure 20: One-tailed t-test results for IBL brain region classification with single modality and MLP classifier with end-to-end fine-tuning. For both modality, NEMO significantly outperforms or matches all baselines.

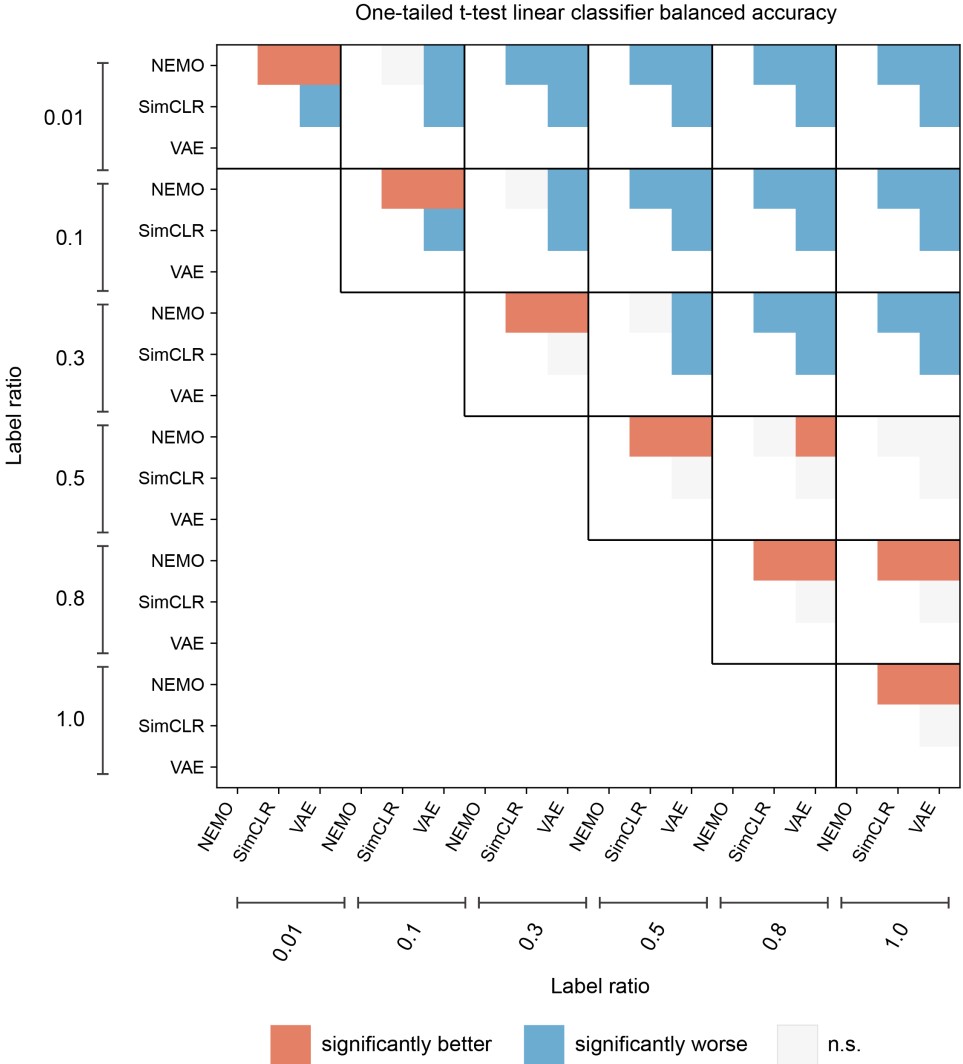

Supplementary Figure 21: One-tailed t-test results on balanced accuracy for IBL brain region classification with linear classifier across different label ratios. For all label ratios, NEMO significantly outperforms all other baselines. With only 50% of the labels, the linear classifier with NEMO either significantly outperforms or shows no significant difference compared to all baselines using 100% of the labels.

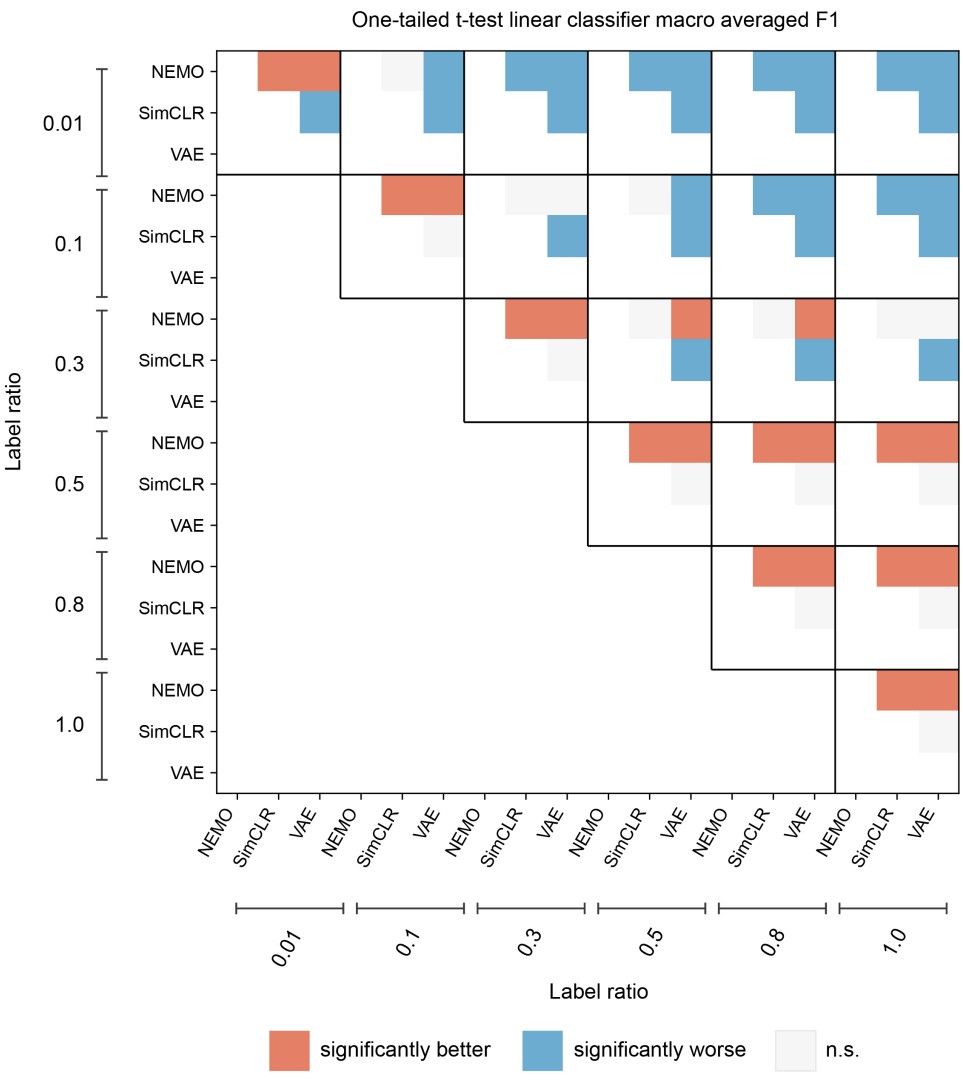

Supplementary Figure 22: One-tailed t-test results on macro averaged F1 score for IBL brain region classification with linear classifier across different label ratios. For all label ratios, NEMO significantly outperforms all other baselines. With only 50% of the labels, the linear classifier with NEMO significantly outperforms all baselines using 100% of the labels.

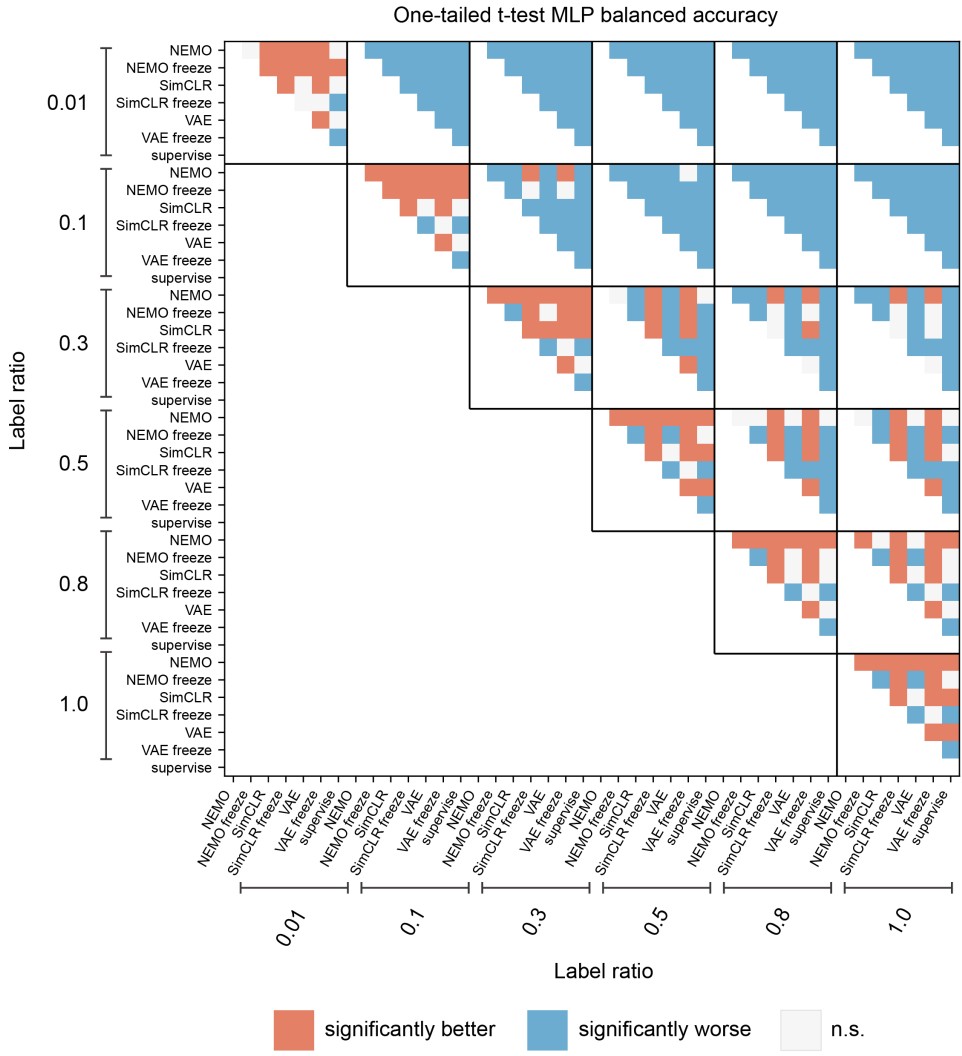

Supplementary Figure 23: One-tailed t-test results on balanced accuracy for IBL brain region classification with MLP classifier across different label ratios. For almost all label ratios (except 0.01), NEMO significantly outperforms all other baselines. With only 80% of the labels, the MLP classifier with NEMO either significantly outperforms or shows no significant difference compared to all baselines using 100% of the labels.

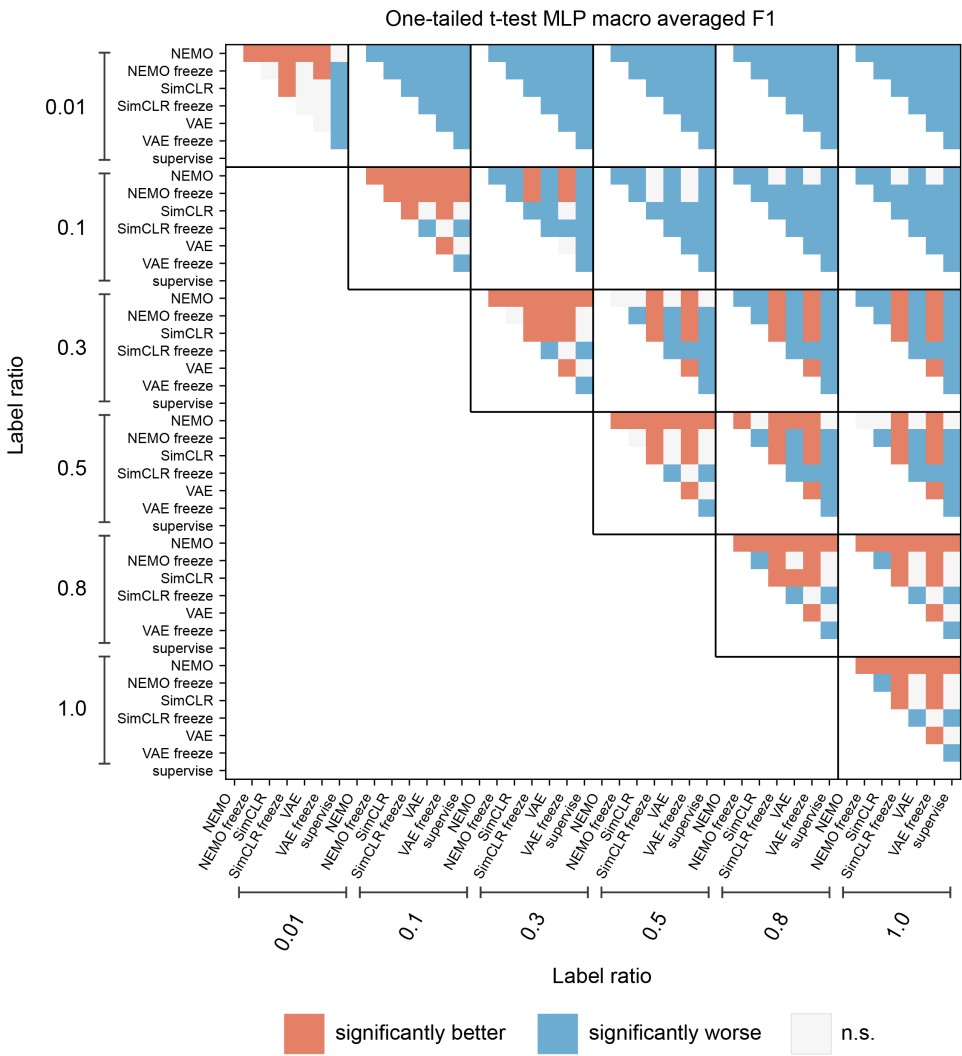

Supplementary Figure 24: One-tailed t-test results on macro F1 score for IBL brain region classification with MLP classifier across different label ratios. For almost all label ratios (except 0.01), NEMO significantly outperforms all other baselines. With only 50% of the labels, the MLP classifier with NEMO either significantly outperforms or shows no significant difference compared to all baselines using 100% of the labels.

