# OpenReview forum: "In vivo cell-type and brain region classification via multimodal contrastive learning"
_ICLR.cc/2025/Conference — ICLR 2025 Spotlight_

### Official Review · Reviewer_a6D4 · 2024-10-18

**Soundness:** 3
**Presentation:** 4
**Contribution:** 2
**Rating:** 6
**Confidence:** 4

**Summary:**

The authors used multimodal (spike waveform + firing pattern) contrastive learning to classify three types of inhibitory neurons (PV/SST/VIP) and ten different brain regions. The proposed NEMO model is based on the widely used CLIP (Contrastive Language-Image Pre-Training) model. NEMO outperforms two previous multimodal models: PhysMAP and VAE.

**Strengths:**

1. The writing is clear, easy to understand, and follows a smooth logical flow, with almost no typos and well-presented figures.

2. The paper provides a thorough review of relevant literature in this field. I have closely followed the cell-type classification area, and all the papers I am aware of (and some I was not) have been accurately cited, except for one (see Weakness). Notably, the authors benchmarked two very recent models that are still in preprint format.

3. This is the second work to use contrastive learning for cell-type classification and the first to combine two modalities.

4. The multimodal approach outperforms single-modal models, which is consistent with previous studies.

**Weaknesses:**

1. In the best-case scenario (cell type classification, L344, Figure 2c), NEMO outperforms VAE by 11%. However, in brain region classification, the improvement is minimal. For example, in Figure 3e, comparing the deep orange (NEMO) to the deep blue (VAE) bars, the difference in balanced accuracy is less than 0.05. Are these differences statistically significant?

2. Joint training shows little to no benefit over independent training. For example, in Figure 5b, comparing the deep orange (NEMO) to the deep violet (SimCLR) bars, the difference in balanced accuracy is around 0.01. Additionally, in Supplementary Table 9, independent NEMO performs either better (0.84 vs. 0.83) or similarly (0.83 vs. 0.84, 0.87 vs. 0.88) to joint NEMO. Are these differences statistically significant?

3. To my knowledge, there is no neuroscience evidence suggesting a strong pairwise correlation between spike waveform and firing patterns. For example, layer 5 pyramidal neurons and cortical PV neurons both fire a large number of spikes (both spontaneously and evoked), but their spike waveforms are broad and narrow, respectively (Cortical connectivity and sensory coding, KD Harris, TD Mrsic-Flogel - Nature, 2013). Additionally, burst firing can be evoked in both excitatory (Chattering cells: superficial pyramidal neurons contributing to the generation of synchronous oscillations in the visual cortex. CM Gray, DA McCormick - Science, 1996) and SST neurons (Somatostatin-expressing neurons in cortical networks, Urban-Ciecko, AL Barth - Nature Reviews Neuroscience, 2016). This is fundamentally different from the relationship between an image and its description in CLIP. In other words, the word "puppy" and an image of a "puppy" represent the same concept, but a broad spike could be associated with either burst or dense firing, depending on whether the neuron is located in layer 2/3 or layer 5.

4. This is the second paper to use contrastive learning for cell-type classification. The first is CEED (Vishnubhotla et al., 2023), which used an unsupervised model (SimCLR) to classify cell types and benchmarked against WaveMap (the single-modality version of PhysMAP). While the CEED work does not significantly compromise the novelty of this study, it should be more clearly acknowledged. The current citation, "Contrastive learning has been applied to raw electrophysiological recordings (Vishnubhotla et al., 2024)," is inappropriate.

**Questions:**

1. In Figure 1a, the text "Neuropixels 1.0" should be replaced with "Neuropixels Ultra" since your VIP/SST/PV data comes from NP Ultra, not NP 1.0, which is used in the IBL dataset for classifying brain regions. Also, please update the inset to show the schematic of NP Ultra instead of NP 1.0.

2. Figure 3b is not mentioned in the text. It could be referenced between L350 and L361.

3. Figure 3c is also not mentioned in the text. It might fit well between L362 and L366.

4. Consider changing the title of Figure 3b and 3c from "unit" to "neuron" to be consistent with the terminology used in the main text.

5. In the PhysMAP paper that you benchmarked, they applied three public datasets from S1, A1, and V1/Hippocampus. Have you tested these datasets? While I am not requesting additional experiments, if you have tested them, it would be helpful to include the results in the supplementary materials.

6. Please specify how you computed the two primary metrics: macro-averaged F1 score and balanced accuracy. Does the first metric equal the unweighted mean of all the per-cell-type F1 scores? Does the second metric equal the unweighted mean of sensitivity (True Positive) and specificity (True Negative)?

---

> ### Author Response · Authors · 2024-11-23
> **Response to Reviewer a6D4**
>
> We thank the reviewer for the detailed and useful review. We hope to address any concerns and questions below.
>
> Weaknesses
> - We thank the reviewer for the suggestion to do significance testing for our results (especially for NEMO’s smaller improvements). We ran a one-sided t-test (with p<.05 significant) for both brain region and cell-type classification. We found that NEMO was significantly better than all baselines (including SimCLR) on brain region classification and also significantly better than all baselines (except SimCLR) on cell-type classification. For cell-type classification, NEMO was significantly better than SimCLR on balanced accuracy using the fine-tuned MLP and the frozen MLP, but not the linear classifier. We also found that NEMO significantly outperforms SimCLR on the f1 score using the frozen MLP, but not the fine-tuned MLP or linear classifier (although the p-value is 0.0505 for the fine-tuned MLP; p < .05 is significant). Please see Supplementary Figures 17-23 with additional details in Supplementary M. Overall, this analysis suggests that NEMO is the strongest model for both brain region and cell-type classification. We hope this alleviates some of the reviewer’s concerns.
> - While we agree that there may not be a perfect 1-to-1 relationship between spike waveform and firing patterns, there is evidence that cell-types can be roughly separated by these two modalities in vivo [1]. Our paper provides further evidence that there is a relatively strong relationship between these two modalities. In light of this, finding the best self-supervised multimodal algorithm will be essential for accurate cell-type classification given the limited opto-tagged labels available.The analogy of the word “puppy” and the image of a “puppy” is interesting, but the relationship between these two are also not strictly 1-to-1 as there are many different types of puppies. Also, multimodal contrastive learning (e.g, CLIP) has a nice property that if the signal is correlated across the two modalities and the noise is independent, then the extracted features can be more informative and generalizable than unimodal contrastive learning where the data augmentations are the only way of learning invariance to the noise (which relies on making assumptions) [2]. We believe this is part of why NEMO does significantly better than SimCLR on the IBL whole-brain dataset which is noisier than the more curated UHD cell-type classification dataset (where performance is more similar).
> - We agree the citation to CEED was inappropriate. We apologize for this oversight and have adjusted the citation to now correctly mention that CEED was utilized to perform cell-type classification. A significant difference in our results is that CEED was never applied to optotagged data and, therefore, all of their cell-typing results are on data with no ground-truth (e.g., their comparison to WaveMAP). Thank you for this correction!
>
> Questions
>
> Figures feedback
> - We have adjusted the figures and text with the feedback provided by the reviewer. We hope this improves the clarity and correctness of the work.
>
> “In the PhysMAP paper that you benchmarked, they applied three public datasets from S1, A1, and V1/Hippocampus. Have you tested these datasets?”
> - We have not tested NEMO the S1, A1, and V1/Hippocampus datasets used in the PhysMAP paper. We appreciate the suggestion by the reviewer and plan to look into these datasets for our future applications. We would also be interested in testing NEMO on the cerebellum dataset from [3] once this becomes publicly available. We believe that the development of a cell-type classification benchmark would also be an exciting future direction given the challenge of preprocessing and utilizing new datasets.
>
> “Please specify how you computed the two primary metrics: macro-averaged F1 score and balanced accuracy.”
> - We have added some clarification for the metrics used in our manuscript. The macro-averaged F1 score is calculated as the unweighted mean of F1 scores for each class (i.e., cell-type or brain region). The balanced accuracy measures the average accuracy per class.
>
> [1] Petersen, Peter C., et al. "CellExplorer: A framework for visualizing and characterizing single neurons." Neuron 109.22 (2021): 3594-3608.
>
> [2] Huang, Wei, et al. "On the Comparison between Multi-modal and Single-modal Contrastive Learning." arXiv preprint arXiv:2411.02837 (2024).
>
> [3] Beau, Maxime, et al. "A deep-learning strategy to identify cell types across species from high-density extracellular recordings." bioRxiv (2024).

---

> > ### Comment · Reviewer_a6D4 · 2024-11-25
> > **Paper Presentation is Excellent**
> >
> > I thank the authors for their statistical tests, which elevate this well-presented paper to an excellent standard. I have updated the presentation score from 3 to 4.
> >
> > I would like to increase my overall score from 6 to 7, though I cannot justify giving it an 8. My primary concern is that the improvement of NEMO over other baselines, while statistically significant, is relatively small. Nevertheless, this is a great paper and is well-qualified for publication.

---

### Official Review · Reviewer_YH4G · 2024-11-01

**Soundness:** 4
**Presentation:** 4
**Contribution:** 3
**Rating:** 8
**Confidence:** 4

**Summary:**

This paper introduces NEMO (Neuronal Embeddings via MultimOdal contrastive learning), a method for classifying neurons by their cell type and brain region location using electrophysiological data. NEMO uses CLIP-like contrastive learning to jointly analyze two types of neural data: the shape of neural electrical signals (waveforms) and patterns of neural activity over time (autocorrelograms).

The authors evaluated NEMO on two datasets: an opto-tagged mouse visual cortex dataset for cell-type classification and the International Brain Laboratory dataset for brain region classification. In comparative experiments, NEMO achieved higher classification accuracy than existing methods including PhysMAP and VAE-based approaches. The authors also demonstrated that using multiple units improved performance, and that the method maintained effectiveness even with limited labeled training data. The paper includes ablation studies examining the importance of joint versus independent learning of the two data modalities.

**Strengths:**

- The paper is well written, the figures are well made, descriptions are generally clear.
- The paper contains extensive additional material for more details.
- The training and experimental setup seems to be carefully chosen and sound.
- Limitations are discussed at the end.

**Weaknesses:**

I only found minor weaknesses.

- Although, the authors did a great job providing necessary details, sometimes the training specifics for the control models and the clustering analyses were a bit too short in my opinion. It would be great if you could provide a bit more detail on this.
- It would be great to see an ablation for the data augmentation to see how important that is (see questions).

**Minor (do not expect you to respond to this)**

- Figure 3 typo in legend “Supervise” instead of “Supervised”
- Supplementary Figure 7 is blurred likely due to plotting settings
- As multi-unit has a particular meaning in neuronscience, I find the wording “multi-unit brain region classification”. I think you mean “multi-neuron” here. Unless I misunderstood and you are really using multi-unit activity, I would change the wording.

**Questions:**

- The VAE training is unclear to me. Do you jointly embed waveforms and autocorrelograms, or do you use two separate encoders/decoders with a shared latent space (which would require cross-decoding, i.e. encode waveform and decoder autocorrelogram)?
- How important are the data augmentations? Can you provide an ablation experiment for that?
- Waveforms and autocorrelograms are two reasonable choices for input modalities. However they are not the only conceivable choices. Have you thought/tried other choices or thought about learning an encoding of spiking activity directly?
- What are the results when you cluster on the latent embeddings directly instead of running UMAP first? How stable is the clustering? I.e. if you train two models of NEMO from different seeds and then cluster neurons, how consistently are two neurons assigned to the same cluster (as measured by adjusted rand index or similar)?
- Can you provide a definition for the modularity index?

---

> ### Author Response · Authors · 2024-11-23
> **Response to Reviewer YH4G**
>
> We thank the reviewer for the useful comments and suggestions. We appreciate that the reviewer found our experiments carefully chosen and sound!
>
> Weaknesses
> - As suggested, we will add additional details for the baselines and clustering analyses in the supplement to improve the clarity of the paper. To this end, we have added a schematic of the encoder architectures in Supplement Figure 2.
> - To test the impact of our different augmentations, we ran two new analyses on the UHD cell-type classification dataset: (1) remove one augmentation and (2) add one augmentation. We have added this result to Supplementary Figure 14 with additional details in Supplementary L.2. Please see the overall response for more details.
> - We agree with the reviewer that multi-unit is an overloaded term for neuroscientists. We have switched single-unit and multi-unit to single-neuron and multi-neuron everywhere in the paper.
> - We have updated figure 3 and the blurry figure in the supplement (Supplementary Figure 7). We hope this fixes the problem!
>
> Questions
>
> “The VAE training is unclear to me.  Do you jointly embed waveforms and autocorrelograms, or do you use two separate encoders/decoders with a shared latent space?”
> - For the VAE baseline, we follow the procedure of [1] which trains a VAE separately on each modality and then concatenates the embeddings before classification.
>
> “How important are the data augmentations? Can you provide an ablation experiment for that?”
> - As mentioned in the above response, we now include an experiment ablating the data augmentations in the overall response.
>
> “Waveforms and autocorrelograms are two reasonable choices for input modalities. However they are not the only conceivable choices. Have you thought/tried other choices or thought about learning an encoding of spiking activity directly?”
> - This is a great question! A couple of ideas we are thinking about is to utilize the peristimulus time histogram (PSTH) or correlations of the neuron with other neurons in the population. We plan to explore these features in future analyses. Encoding the spiking activity directly is another interesting idea although this would require a much more complicated neural network architecture.
>
> “What are the results when you cluster on the latent embeddings directly instead of running UMAP first? How stable is the clustering?”
> - To quantify the stability of the NEMO clustering result, we re-ran our clustering pipeline with multiple random seeds. Although there is shared structure between clusterings, we found that there was variability with different random seeds (see Supplementary Figure 16). We expect this to be the case because clustering the full brain dataset leads to a very coarse clustering when, in reality, there is much more variability per region. We should be cautious over interpreting these clusters as this is more of an exploratory analysis. We will add this caveat to the paper.
>
> “Can you provide a definition for the modularity index?”
> - The modularity index is a hyperparameter in the Louvain clustering that quantifies the relative density of edges within communities compared to the edges between communities. It ranges from −0.5 (indicating non-modular clustering) to 1 (indicating fully modular clustering). Tuning this parameter will affect how the different nodes are grouped together in the network.
>
> [1] Beau, Maxime, et al. "A deep-learning strategy to identify cell types across species from high-density extracellular recordings." bioRxiv (2024).

---

> > ### Comment · Reviewer_YH4G · 2024-11-23
> > **Thanks**
> >
> > Thanks for your response and the additional analyses. I still think it's a good paper: I kept my score and increase my confidence.

---

### Official Review · Reviewer_vN2L · 2024-11-03

**Soundness:** 3
**Presentation:** 3
**Contribution:** 3
**Rating:** 8
**Confidence:** 4

**Summary:**

The authors proposed a multimodal contrastive learning approach for joint modeling of extracellular action potential data and spiking activity data. The proposed approach can be fine-tuned for different downstream tasks, including in vivo cell-type and brain region classification. More specifically, the author applied the classic contrastive learning framework established in CLIP on extracellular action potential data and spiking activity data. Although the theoretical innovation is relatively limited, the authors made the earliest efforts (as far as I know) to apply contrastive learning for joint modeling of extracellular action potential data and spiking activity data.

**Strengths:**

As mentioned above, this study is among the earliest efforts to apply contrastive learning for joint modeling of extracellular action potential data and spiking activity data. Most key components in the proposed analytical framework are fetched from previous work (e.g., CLIP contrastive learning and the ACG encoder), increasing the reproducibility of the study. In addition, the study was well presented and easy to follow.

**Weaknesses:**

As mentioned above, the theoretically contribution of the study is relatively limited. The readers may expect to see some components that are specifically designed with consideration for the unique characteristics of the data and the problem being  addressed.

**Questions:**

1.The authors emphasized “in vivo”, however, are there any results about the computational efficiency of the NEMO model that can support it? Does it support real-time processing?

2.I would expect a validation of the data augmentation strategy. It is understandable that the construction of ACG images is computationally expensive. But the authors are encouraged to validate the data augmentation strategy adopted in the study, augmentations directly for the ACG images, is reasonable by showing a couple examples.

3.The authors compared NEMO with the VAE-based method. However, in my opinion, it seems that a critical comparison is missing: the comparison between naive models, specifically, NEMO without fine-tuning and the VAE-based method without fine-tuning. This comparison would highlight the representational power of the two methods .

---

> ### Author Response · Authors · 2024-11-23
> **Reponse to Reviewer vN2L**
>
> We thank the reviewer for their feedback and we hope to address some concerns below
>
> Weaknesses
> - Novelty: While we make good use of previous work, we also introduce novel data augmentations and brain classification approaches in our paper. We are also, to our knowledge, the first application of multimodal contrastive learning for either brain region or cell-type classification. Overall, we think NEMO will be a valuable machine learning tool for systems neuroscientists.
>
> Questions
>
> “The authors emphasized “in vivo”, however, are there any results about the computational efficiency of the NEMO model that can support it? Does it support real-time processing?”
> - To clarify, by “in vivo” we mean that the recordings we analyze are from the brain in its awake and behaving state. We clarify this so that the reader understands that we are not doing any “in vitro” analysis (i.e., outside the brain). This is similar terminology as used in other papers including [1]. We do not make any claims about real-time processing and we are happy to clarify this in the final version of the paper.
>
> “... The authors are encouraged to validate the data augmentation strategy adopted in the study, augmentations directly for the ACG images, is reasonable by showing a couple examples.”
> - To validate the impact of our different augmentations, we ran two new analyses on the UHD cell-type classification dataset: (1) remove one augmentation and (2) add one augmentation. We have added this result to Supplementary Figure 14 with additional details in Supplementary L.2. We also show example augmentations in Supplementary Figure 3. Please see the overall response for more details. We thank the reviewer for this suggestion.
>
> “The authors compared NEMO with the VAE-based method. However, in my opinion, it seems that a critical comparison is missing: the comparison between naive models”
> - We agree that a comparison of naive models is important. We actually do quantify the performance of the naive models in our paper. We have three evaluation schemes: linear classification using the embeddings (i.e., linear classifier), MLP classification using the embeddings (i.e. frozen MLP), and full end-to-end fine-tuning (fine-tuned MLP). Both the linear classifier and frozen MLP evaluate the performance of each model without any additional fine-tuning of the embedding method (e..g, NEMO, VAEs, etc.). We hope this clarification addresses the reviewer’s concern.
>
> [1] International Brain Laboratory, et al. "Reproducibility of in-vivo electrophysiological measurements in mice." bioRxiv (2022): 2022-05.

---

> > ### Comment · Reviewer_vN2L · 2024-11-25
> >
> > I would like to thank the reviewer for their effort in addressing my concerns. Most of my concerns have been adequately addressed.

---

### Official Review · Reviewer_iQCh · 2024-11-04

**Soundness:** 4
**Presentation:** 4
**Contribution:** 4
**Rating:** 8
**Confidence:** 3

**Summary:**

This paper presents a novel application of contrastive learning to solve two important problems in systems neuroscience by incorporating just two electrophysiological recording modalities: spiking activity and extracellular action potentials (EAPs). The authors developed a new framework called Neuronal Embeddings via Multimodal Contrastive Learning (NEMO) by combining the well-established Contrastive Language-Image Pretraining (CLIP) framework with task-specific data augmentations and encoders. The authors demonstrate the multimodality as well as the power and utility of NEMO by evaluating its performance on two very different downstream tasks:

1.	cell-type classification among parvalbumin (PV), somatostatin (SST), and vasoactive intestinal peptide (VIP) inhibitory interneurons using opto-tagged Neuropixels Ultra (NP Ultra) recordings from the mouse visual cortex, and

2.	brain-region classification among 10 broad areas using the public International Brain Laboratory (IBL) brain-wide map data set.

This paper’s novelty mainly stems from the utilization of a paired data set that combines an autocorrelogram (ACG) image of every neuron’s spiking activity and a waveform template of the neurons’ EAPs, and from the application of two separate encoders for the aforementioned two modalities. In both cell-type and brain-region classification tasks, the authors show that NEMO outperforms the state-of-the-art multimodal cell-type embedding methods including PhysMAP and VAE-based methods as well as a fully supervised method in terms of balanced accuracies and macro-averaged F1 scores with minimal fine-tuning.

**Strengths:**

This is a very well-written paper with clear organization of the figures and sound presentation of the data sets used, methods applied, and results obtained. The authors apply a successful contrastive learning method to develop a new framework that outperforms comparable state-of-the-art methods in both the cell-type classification task and the less widely-explored brain-region classification task. The task relies on two electrophysiological recording modalities: spiking activity and EAPs, which are more accessible, and the decent classification performances come with minimal fine-tuning. NEMO is able to differentiate between VIP and SST cells, which is highly valued in systems neuroscience, and its ability to yield data embeddings that lead to separable classification regions is impressive. The method described in this paper will be particularly helpful and useful to systems neuroscientists interested in applying this technique with the goal of decoding the neural circuitry underlying multiple biological functions.

**Weaknesses:**

Detailed graphical representations of the architectures used in the authors’ method may help the readers understand the details of NEMO better. A more comprehensive description, such as including explanations of 10D and 500D in the VAE baseline versions’ latent spaces of the encoder architectures, would have further aided clarity to the method’s explanations.

The authors restrict the number of example neurons, whose recorded activities are inputted to the overall architecture, to five neurons without an explanation of why the input neuron number was kept low. Providing a rationale for limiting the number of input neurons to five as well as an explanation of how the results of the authors' method change with varying input neuron number would be greatly appreciated.

The authors state that they fixed the hyperparameters for all methods across the two data sets used in the experiments due to the limited number of labels for evaluation for the cell-type classification task. It is unclear whether this choice led to fair performance comparisons among the state-of-the-art methods. It would help to know that separate hyperparameter optimization among the different methods and data sets would not yield different results. Providing a brief analysis on the sensitivity of the results and the overall performance to variations in specific hyperparameters will notably strengthen the claims made in this paper.

Minor comments:

There seems to be a citation error or a missing preposition in Section 1 when the authors cite IBL et al.

Radford et al. 2021 to Radford et al., 2021 in Section 4

Table 6 is non-existent in Section 6.1. The authors may be referring to Table 1.

There seems to be a citation error or a missing preposition in Section 6.3 when the authors cite Chen et al. (2020).

Figure 5 (b) caption's word "then" should be changed to "than."

In Section 7, the phrase "should also be useful these down-stream tasks" should be changed to "should also be useful in these down-stream tasks."

**Questions:**

The waveform template is restricted to one channel with maximal amplitude and multi-channel template results are shown in Supplement E. What exactly is the definition of a channel in this context?

Why was additive Gaussian noise chosen as the sole data augmentation? Can a brief rationale for this specific choice be included in the paper? The authors demonstrate that adding two template augmentations: amplitude jitter and electrode dropout does not significantly improve the performance of the two downstream classification tasks. How do other data augmentation types impact the performance and results?

---

> ### Author Response · Authors · 2024-11-23
> **Response to Reviewer iQCh**
>
> We thank the reviewer for the thoughtful comments and suggestions. We appreciate that the reviewer believes that NEMO will be useful for systems neuroscientists; This is our hope as well!
>
> Additional experiments and figures
> - We agree that the description of the architectures was too limited so we have added a schematic of the encoder architectures in Supplement Figure 2.
> - We chose 5 neurons per depth as an initial starting point to evaluate the contribution of multi-neuron ensembling. We agree with the reviewer that this analysis could be improved so we ran an additional experiment where we vary the number of neurons from 3 - 25 for brain region classification. Please see the overall response for more details.
> - To understand the sensitivity of NEMO and the VAE baseline to hyperparameters, we are currently running a grid search over the learning rate, dropout, and embedding dimension for the UHD cell-type classification dataset. We plan to add this to the supplement once it finishes running and we will update the rebuttal accordingly.
> - To test the impact of our different augmentations, we ran two new analyses on the UHD cell-type classification dataset: (1) remove one augmentation and (2) add one augmentation. We have added this result to Supplementary Figure 14 with additional details in Supplementary L.2. Please see the overall response for more details.
>
> Questions
>
> “What exactly is the definition of a channel in this context?”
> - The definition of a channel in this context is a single electrode. So when we restrict the template to one channel, we are restricting the template to be the waveform recorded on the single largest amplitude electrode.
>
> “Why was additive Gaussian noise chosen as the sole data augmentation?”
> - We used gaussian noise for single-channel templates to help compensate for low spike count templates being noisier. We could not think of any additional nuisance variables for single-channel templates. For multi-channel templates, we introduced electrode dropout and per-channel amplitude jitter which lead to improvements for NEMO on both brain region and cell-type classification (please see Supplement E). We will add this explanation to the main text.
>
> “How do other data augmentation types impact the performance and results?”
> - To test the impact of our different augmentations, we ran two new analyses on the UHD cell-type classification dataset: (1) remove one augmentation and (2) add one augmentation. We thank the reviewer for the suggestion. Please see the overall response for more details.

---

### Author Response · Authors · 2024-11-23
**Overall response to reviewers**

We thank the reviewers for all the thoughtful feedback and suggestions. We are happy that the reviewers had a lot of positive feedback for the work, including:
- “This is a very well-written paper with clear organization of the figures and sound presentation of the data sets used, methods applied, and results obtained.” (iQCh); “the study was well presented and easy to follow.” (vN2L)
- “The training and experimental setup seems to be carefully chosen and sound.” (YH4G)
- “The multimodal approach outperforms single-modal models, which is consistent with previous studies.”
- “NEMO is able to differentiate between VIP and SST cells, which is highly valued in systems neuroscience, and its ability to yield data embeddings that lead to separable classification regions is impressive.” (iQCh)

Based upon the reviewer comments and questions, we ran four new experiments and added new supplements detailed below:

NEMO data augmentation ablations
- To evaluate the impact of our proposed data augmentations, we ran two new analyses on the UHD cell-type classification dataset: (1) remove one augmentation and (2) add one augmentation. For the remove one augmentation experiment, we remove one augmentation and keep the rest. For the add 1 augmentation experiment, we only use one augmentation and remove the rest. We compare these new NEMO models to a version of NEMO with all augmentations and a version of NEMO without any augmentations using the linear classifier to predict cell-types. The results of this analysis can be seen in Supplementary Figure 14 with additional details in Supplementary L.2. As can be seen, the no augmentations model performs noticeably worse than with the models with augmentations. Additive Gaussian noise for the ACGs is especially helpful. For other augmentations, while the combined effect is large, the individual contribution is smaller. For the final version of the paper, we plan to ablate multiple augmentations at once to see which combinations are most important for the final performance of NEMO.

Significance testing NEMO vs. baselines
- As recommended by reviewer a6D4, we performed significance testing of the improvement of NEMO over all baseline models (VAE, SimCLR, supervised, PhysMAP) for all experiments using a one-tailed t-test (with p < .05 significant). (1) Brain region classification: For both the linear and the MLP classifiers, we find that NEMO significantly outperforms all baselines across all label ratios (except supervised vs. NEMO for 1% of the labels). We also find that NEMO significantly outperforms all baselines (which use 100% of the labels) with only 80% of the labels. (2) Cell-type classification: For both the linear and MLP classifiers, we find that NEMO significantly outperforms all baselines (except SimCLR). We found that NEMO significantly outperforms SimCLR on balanced accuracy using the fine-tuned MLP and the frozen MLP, but not the linear classifier. We also found that NEMO significantly outperforms SimCLR on the f1 score using the frozen MLP, but not the fine-tuned MLP or linear classifier (although the p-value is 0.0505 for the fine-tuned MLP; p < .05 is significant). The results of this analysis can be seen in Supplementary Figures 17-23 with additional details in Supplementary M. Overall, this analysis suggests that NEMO is the strongest model for both brain region and cell-type classification.

Multi-neuron brain region classification with 3 - 25 neurons
- In our paper, we originally used 5 neurons for all multi-neuron brain region classification results. To evaluate the impact of this choice, we re-ran multi-neuron brain region classification with NEMO using varying numbers of neurons from (i.e., 3 - 25). The results of this analysis are shown in Supplementary Figure 15 with additional details in Supplementary L.3. As can be seen, increasing the number of neurons marginally improves the performance of brain region classification with NEMO with saturation at 25 neurons.

Hyperparameter sensitivity analysis
- To evaluate the hyperparameter sensitivity of NEMO, we are currently performing a grid search over the learning rate, dropout, and embedding size for NEMO and the VAE-based method. We will include this new result in the Supplementary once it has finished running. We will also update our rebuttal accordingly.

---

> ### Author Response · Authors · 2024-11-25
> **Update to overall response**
>
> We have finished running the hyperparameter sensitivity analysis. The results for this analysis can be found in Supplement L, Supplementary Figure 13, and Supplementary Table 4. As can be seen, NEMO is robust to the range of hyperparameters we tested: the average performance over the tested hyperparameters is the same as the default NEMO model reported throughout the paper. The VAE is also robust to the range of hyperparameters we tested. We also found that the learning rate we used for cell-type classification for the default VAE was lower than the original learning rate from [1] (we adjusted this to improve reconstruction) and that a larger learning rate improves downstream performance. We have adjusted this parameter and updated Figure 2 and Table 1. NEMO still substantially outperforms the VAE baselines and the linear NEMO model is still significantly better than the end-to-end fine-tuned VAE. We thank the reviewers for this suggested analysis.
>
> [1] Beau, Maxime, et al. "A deep-learning strategy to identify cell types across species from high-density extracellular recordings." bioRxiv (2024).

---

### Comment · Area_Chair_hNox · 2024-11-27
**Reminder: Last day for author feedback**

This is a reminder that today is the last day allotted for author feedback. If there are any more last minute comments, please send them by today.

---

### Meta-Review · Area_Chair_hNox · 2024-12-19

**Metareview:**

The authors proposed an application of contrastive learning for spiking activity and extracellular action potentials (EAPs). The proposed Neuronal Embeddings via Multimodal Contrastive Learning (NEMO) utilizes Contrastive Language-Image Pretraining (CLIP) task-specific data augmentations and encoders. Although the theoretical innovation is relatively limited, to the best of the referees' knowledge, this is the second work to use contrastive learning for cell-type classification and the first to combine two modalities (extracellular action potential data and spiking activity data). We have read the referee reports and the author responses. The primary concerns include restricted limited algorithmic innovation, number of neurons, and experiments (e.g., ablation for the data augmentation, significance testing, sensitivity to hyperparameters, increasing number of neurons for brain region classification). The authors were able to address most of the issues raised. We believe the comments and suggestions from the referees have improved the quality of the manuscript and encourage the authors to incorporate points discussed into the next revision.

**Additional Comments On Reviewer Discussion:**

The primary concerns include restricted limited algorithmic innovation, number of neurons, and experiments (e.g., ablation for the data augmentation, significance testing, sensitivity to hyperparameters, increasing number of neurons for brain region classification). The authors were able to address most of the issues raised.

---

### Decision · Program_Chairs · 2025-01-22

Accept (Spotlight)